# Strained few-layer MoS$_2$ with atomic copper and selectively exposed in-plane sulfur vacancies for CO$_2$ hydrogenation to methanol

Shenghui Zhou[1,2], Wenrui Ma [1], Uzma Anjum[1], Mohammadreza Kosari [1,3], Shibo Xi[3], Sergey M. Kozlov [1] ✉ & Hua Chun Zeng [1,2] ✉

In-plane sulfur vacancies (Sv) in molybdenum disulfide (MoS$_2$) were newly unveiled for CO$_2$ hydrogenation to methanol, whereas edge Sv were found to facilitate methane formation. Thus, selective exposure and activation of basal plane is crucial for methanol synthesis. Here, we report a mesoporous silica-encapsulated MoS$_2$ catalysts with fullerene-like structure and atomic copper (Cu/MoS$_2$@SiO$_2$). The main approach is based on a physically constrained topologic conversion of molybdenum dioxide (MoO$_2$) to MoS$_2$ within silica. The spherical curvature enables the generation of strain and Sv in inert basal plane. More importantly, fullerene-like structure of few-layer MoS$_2$ can selectively expose in-plane Sv and reduce the exposure of edge Sv. After promotion by atomic copper, the resultant Cu/MoS$_2$@SiO$_2$ exhibits stable specific methanol yield of 6.11 mol$_{MeOH}$ mol$_{Mo}$$^{-1}$ h$^{-1}$ with methanol selectivity of 72.5% at 260 °C, much superior to its counterparts lacking the fullerene-like structure and copper decoration. The reaction mechanism and promoting role of copper are investigated by in-situ DRIFTS and in-situ XAS. Theoretical calculations demonstrate that the compressive strain facilitates Sv formation and CO$_2$ hydrogenation, while tensile strain accelerates the regeneration of active sites, rationalizing the critical role of strain.

The excessive anthropogenic CO$_2$ emission in the atmosphere is driving up global warming and serious climate and environmental issues[1–3]. To alleviate such dilemma, catalytic hydrogenation of CO$_2$ with renewable H$_2$ to produce clean liquid fuels and value-added chemicals, so-called "liquid sunshine", is perceived to be a promising sustainable approach to simultaneously mitigate greenhouse effect and relieve energy shortage in the 21st century[3–6]. Particularly, methanol synthesis from CO$_2$ has attracted increasing attention in recent years. This C$_1$ alcohol is not only considered a viable alternative fuel with a high energy density but also opined as a feedstock for conversion into olefins, gasoline, and other downstream bulk chemicals, thus forming a backbone of "methanol economy"[7–10]. In the last few decades, extensive research efforts have been dedicated to developing efficient catalysts for methanol synthesis from CO$_2$ hydrogenation; such catalysts include Cu-metal oxides (Cu/ZnO/Al$_2$O$_3$, Cu/ZrO$_2$)[11–13], In$_2$O$_3$-based oxide[14,15], solid solution (ZnO/ZrO$_2$[16], GaZrO$_x$[17], In$_2$O$_3$/ZrO$_2$[18]), metal alloys (NiGa[19], MnCo[20], PdZn[21], PdIn[22]), and Mo-containing solids (MoP[23], β-Mo$_2$C[24], MoS$_2$[25], etc.).

[1]Department of Chemical and Biomolecular Engineering, College of Design and Engineering, National University of Singapore, Singapore 119260, Singapore. [2]The Cambridge Centre for Advanced Research and Education in Singapore, 1 CREATE Way, Singapore 138602, Singapore. [3]Institute of Sustainability for Chemicals, Energy and Environment (ISCE2), Agency for Science, Technology and Research (A*STAR), 1 Pesek Road, Jurong Island, Singapore 627833, Singapore. ✉e-mail: sergey.kozlov@nus.edu.sg; chezhc@nus.edu.sg

Among these catalysts, few-layer $MoS_2$ nanosheets, which were newly unveiled for $CO_2$ hydrogenation in 2021, had attracted much attention because they worked at relatively low reaction temperatures with satisfactory catalytic performance and stability for methanol synthesis[25]. $MoS_2$ is a prototypical two-dimensional layered transition metal dichalcogenide, consisting of three atomic layers, S−Mo−S, which then form stacked multiple-layered solids through van der Waals forces. Unlike common metal oxide catalysts, $MoS_2$ also features more complex structures which have a profound impact on selective methanol synthesis. It has three polytypes (1 T, 2H, and 3 R), various layered structures (multilayer, few-layer, and single-layer), two types of active sites (in-plane and edge sites), and two types of vacancies (Mo vacancy and S vacancy)[26]. Only the few-layered 2H-phase $MoS_2$ with sufficiently exposed in-plane S vacancies (Sv) can catalyze methanol synthesis, while $MoS_2$ with abundant edge S vacancies mainly promotes methane production[25]. 1 T and 3 R phase $MoS_2$ cannot effectively catalyze this reaction. Therefore, although the synthesis of common $MoS_2$ is not complicated, conventional synthetic $MoS_2$ were generally multilayer/thick-layer structures with inert basal planes and randomly exposed Sv, exhibiting poor performance for methanol synthesis[25]. In this regard, enhancing the catalytic performance of $MoS_2$ for selective $CO_2$ hydrogenation to methanol can be achieved through two promising strategies. Firstly, improving the dispersity of 2H-phase $MoS_2$ slabs and reducing the degree of layer stacking in $MoS_2$ show potential in promoting catalytic activity. Secondly, an effective approach involves selectively exposing and activating the inert basal plane while simultaneously minimizing the generation and exposure of edge catalytic sites in $MoS_2$ catalysts.

Recently, the application of strain engineering in two-dimensional transition metal dichalcogenides has shown promise as an effective strategy for activating their basal plane[28,29]. Remarkably, the introduction of strain into $MoS_2$ serves a dual purpose: it not only allows for the regulation of the electronic structure of active sites but also facilitates the generation of Sv, thereby establishing a favorable environment for the targeted reaction[29]. Generally, strain can be created by the inheritance of wrinkled elastomeric substrates or patterned rigid substrates through mechanical transfer techniques[26,27,29-31]. However, most of these methods are complicated to adopt and difficult to scale up; they are generally expensive and require specialized equipment[32,33]. In addition, these preparative processes are mainly based on electrochemical methods and the obtained form of $MoS_2$ usually features multilayer stacks, thus not suitable for high-pressure heterogeneous gas-solid systems. On the other hand, the previously developed $MoS_2$ catalysts can only randomly expose in-plane Sv and edge Sv. Considering that in-plane Sv catalyzes methanol synthesis while edge Sv promotes methane production[25], herein we hypothesize that if $MoS_2$ features an ideal few-layer fullerene-like structure with only in-plane Sv and no or least edge Sv, this type of layered catalysts will then selectively catalyze $CO_2$ to methanol with CO as the only byproduct. Nevertheless, the precisely controllable synthesis of strained 2H-phase $MoS_2$ with both few-layer and fullerene-like structural features for selective $CO_2$ hydrogenation has remained a formidable experimental challenge and has not yet been demonstrated.

In recent years, our team has conducted a range of research on silica-encapsulated nanostructures to address unsolved issues in some well-established fields through tailormade catalytic nanomaterials and/or nanocomposites[34-36]. For instance, microporous silica - encapsulated $Pd/FeO_x$ was prepared to confirm the Suzuki−Miyaura cross-coupling reaction mechanism[37]. Mesoporous silica - encapsulated metal−organic frameworks were constructed to demonstrate significantly enhanced mechanical properties with better stability in catalysis[38]. Mesoporous silica - encapsulated $MoO_2$ solid precursor was built to achieve a targeted synthesis of highly active silicomolybdic acid catalysts[39]. Based on our previous research progress on silica encapsulating materials and the precisely controllable synthesis of

ultrafine $MoO_2$ nanosphere over the years, in this study, we developed a synthetic protocol for fabricating fullerene-like $MoS_2$ hollow spheres encapsulated inside a mesoporous silica shell. Physically constrained topological sulfidation of ultrafine $MoO_2$ within hollow mesoporous silica sphere enables the generation of few-layer and spherical $MoS_2$ with in-plane strain and Sv. Importantly, bent $MoS_2$ nanosheets can selectively expose in-plane Sv, which is conducive to methanol synthesis. By further anchoring atomic Cu onto the strained $MoS_2$, the resultant $Cu/MoS_2@SiO_2$ delivered an extraordinary methanol selectivity and specific methanol yield, which markedly surpassed commercial $Cu/ZnO/Al_2O_3$ and previously reported $MoS_2$-based catalysts. Finally, the investigation of the reaction mechanism and the enhancing influence of Cu was conducted using in situ DRIFTS and in situ XAS techniques; DFT calculations were also employed to elucidate the crucial role of Cu and strain in our $Cu/MoS_2@SiO_2$ system for $CO_2$ hydrogenation.

## Results

### Synthesis and characterizations of catalysts

Figure 1a schematically illustrates our synthetic protocol for the preparations of $MoO_2$, $MoO_2@SiO_2$, $MoS_2@SiO_2$, and $Cu/MoS_2@SiO_2$ samples. Briefly, uniform $MoO_2$ nanocores were first prepared by a simple hydrothermal method with polyvinylpyrrolidone (PVP) as the capping agent in a water-ethanol cosolvent, as previously developed by our group[39,40]. The $MoO_2$ nanocores were utilized as a template for the synthesis of $MoO_2@SiO_2$ through the deposition of a uniform silica shell in a water-methanol cosolvent. The incorporation of cetyltrimethylammonium chloride (CTAC) surfactant during this process led to the formation of perpendicular mesoporous channels within the shell structure. After that, the $MoO_2$ core of $MoO_2@SiO_2$ was sulfurized to $MoS_2$ with thioacetamide (TAA) as the sulfide source in hydrothermal conditions. Subsequently, copper was introduced into the $MoS_2@SiO_2$ structure through impregnation, resulting in $Cu/MoS_2@SiO_2$. It was worth noting that $MoO_2$ nanocores could also be directly transformed into $MoS_2$ nanoparticles ($MoS_2$-NPs) without requiring the intermediate step of silica encapsulation (Fig. 1b). In addition, for catalytic performance comparison, multilayered $MoS_2$-HT could also be synthesized by traditional hydrothermal method using ammonium heptamolybdate tetrahydrate (AMT) as Mo source and TAA as sulfur source (Fig. 1c). The morphologies and structures of different $MoO_2$ and $MoS_2$ nanocomposites were revealed and analyzed by field-emission scanning electron microscopy (FESEM) and transmission electron microscopy (TEM). As shown in Fig. 2a–c, $MoO_2$ nanocores exhibit a very uniform irregular spherical structure with an average diameter of approximately 47 nm. To optimize the formation conditions, the synthesis parameters, including the quantities of AMT, PVP, and the type of solvent were systematically explored. The results show that the particle size of $MoO_2$ nanocores can be precisely controlled, ranging from 30 nm to 150 nm by simply changing the amount of PVP or AMT added (Supplementary Figs. 1, 2). Besides, as capping agent and reducing agent, both PVP and ethanol are indispensable for the synthesis of ultrafine $MoO_2$ (Supplementary Fig. 3). It is worth mentioning that precise control of $MoO_2$ particle size plays a decisive role in the thickness of synthesized $MoS_2$ afterwards. After the silica deposition process, representative FESEM images reveal that $MoO_2@SiO_2$ maintains the uniform spherical morphology of $MoO_2$ (Fig. 2d). The TEM and high-resolution TEM (HRTEM) images (Fig. 2e–f) indicate that all $MoO_2$ cores with an average size of ~47 nm are uniformly wrapped by a well-defined $SiO_2$ shell with an average thickness of around 53 nm. It is worth noting that most of $MoO_2@SiO_2$ consist of a single $MoO_2$ nanocore, with occasional exception of two or more nanocores (Supplementary Fig. 4a). Furthermore, high-angle annular dark-field scanning TEM images (HAADF-STEM, Fig. 2g), corresponding energy-dispersive X-ray spectroscopy (EDS) mapping images (Fig. 2h) and elemental line scan profiles (Fig. 2i) clearly reveal

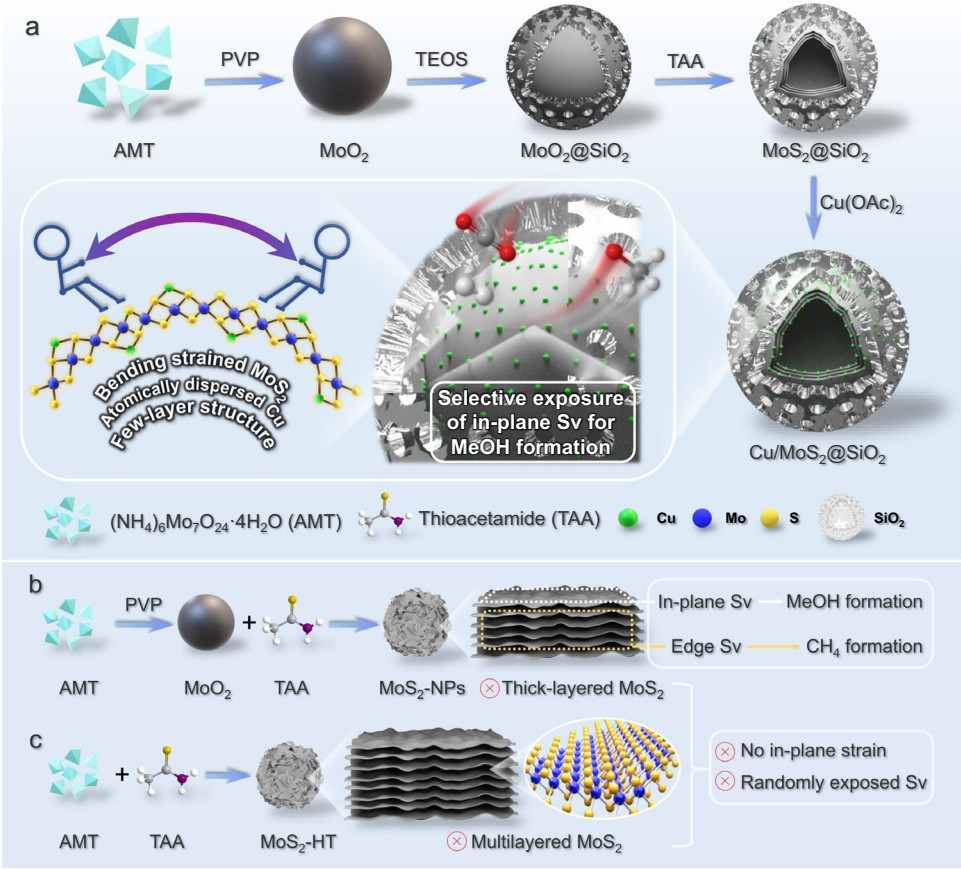

**Fig. 1 | Schematic illustrations of targeted syntheses of different samples. a** $MoO_2$, $MoO_2@SiO_2$, $MoS_2@SiO_2$ and $Cu/MoS_2@SiO_2$. **b** $MoS_2$-NPs. **c** $MoS_2$-HT.

that the core−shell nanostructure of $MoO_2@SiO_2$ and the existence of Mo, O and Si elements. Among them, the Si signal is predominantly located in the outer shell, while Mo signal mainly situate in the inner core, further confirming all the $MoO_2$ cores are well-encapsulated by $SiO_2$ shell.

To transform the $MoO_2$ core to $MoS_2$, TAA was utilized as the sulfiding agent. Under hydrothermal conditions, TAA reacted with $MoO_2$, giving rise to the formation of $MoS_2$. The FESEM image (Fig. 3a1) shows that the core-shell spheres retain their initial size and exhibit a uniformly smooth outer shell, while the TEM images (Fig. 3a2, a3) reveal that the inner $MoO_2$ core become a fullerene-like hollow structure comprising 2–4 layers of spherical $MoS_2$. Moreover, the HRTEM image (Fig. 3a4) reveals a characteristic lamellar structure with well-resolved $d$-spacing of 0.65 nm, which correspond to the lattice fringes of $d_{002}$ of 2H·$MoS_2$ phase (JCPDS No. 37-1492). High-angle annular dark-field scanning transmission electron microscopy with an energy-dispersive X-ray spectroscopy (HAADF-STEM-EDS) elemental mapping (Fig. 3a5) and corresponding elemental line scanning profiles (Supplementary Fig. 5a) confirm the existence of Mo, S, Si and O elements. Specifically, the Si signal is primarily located in the silica shell, while the Mo and S signals are mainly concentrated in the core region, further confirming the core-shell hollow spheres of $MoS_2@SiO_2$.

Systematic exploration of hydrothermal parameters, including reaction temperature, time, types of solvents, and sulfur source, was also pursued to investigate the formative evolution of $MoS_2@SiO_2$ with the hollow core of $MoS_2$. The transformation of $MoO_2@SiO_2$ to $MoS_2@SiO_2$ was found to necessitate a minimum temperature of 200 °C for 24 hours. At low temperatures of 160 and 180 °C, $MoO_2$ was found difficult to be sulfidized and it partially dissolved in aqueous solutions, leading to evacuation of $MoO_2$ nanocore (Supplementary Fig. 6). When the sulfidation reaction time was shortened, TEM images

show that the outer surface of $MoO_2$ is partially sulfidized while the central part is not (Supplementary Fig. 7). In addition, when the sulfur source is replaced with thiourea (Supplementary Fig. 8a), we can also obtain bending strained $MoS_2@SiO_2$. However, when the solvent is replaced by ethanol, only a portion of $MoS_2$ in $MoS_2@SiO_2$ feature spherical curvature (Supplementary Fig. 8b). These findings suggest that fast release and long-lasting supply of $S^{2-}$ ions in water from TAA play a pivotal role in the formation of the strained $MoS_2$ hollow spheres.

The as-prepared $MoS_2@SiO_2$ was further decorated with Cu species. The FESEM and TEM images (Fig. 3b1– 3b4) show that the silica - encapsulated core-shell spherical morphology and confined few-layer fullerene-like $MoS_2$ structure are well preserved after the introduction of Cu. Nonetheless, identification of actual localization of the loaded Cu species within the same image is not achievable, suggesting that the loaded Cu is present in an extremely small form, potentially in the form of ultrafine clusters or even single atoms. On the other hand, the Si signal is predominantly detected in the shell region, while Cu, Mo, and S signals mainly situated in the interior space, as evidenced by our investigations through HAADF-STEM-EDS elemental mapping (Fig. 3b5) and corresponding elemental line scanning profiles (Supplementary Fig. 5b), further revealing the copper is localized in the $MoS_2$ phase. This phenomenon can potentially arise from the contraction of the metal solution from the external region of the hollow mesoporous silica shell towards the interior during the drying process following impregnation, which aligns with our previously reported findings[41]. In addition, the disparity observed between $mSiO_2$ and $MoS_2$ can be attributed to the stronger surface affinity of unsaturated sulfur in $MoS_2$ towards $Cu^{2+}$[42]. In addition, ICP-OES results show that the $MoS_2$ and Cu contents in the $Cu/MoS_2@SiO_2$ catalyst are 34% and 1.45%, respectively (Supplementary Table 1). For comparison, direct

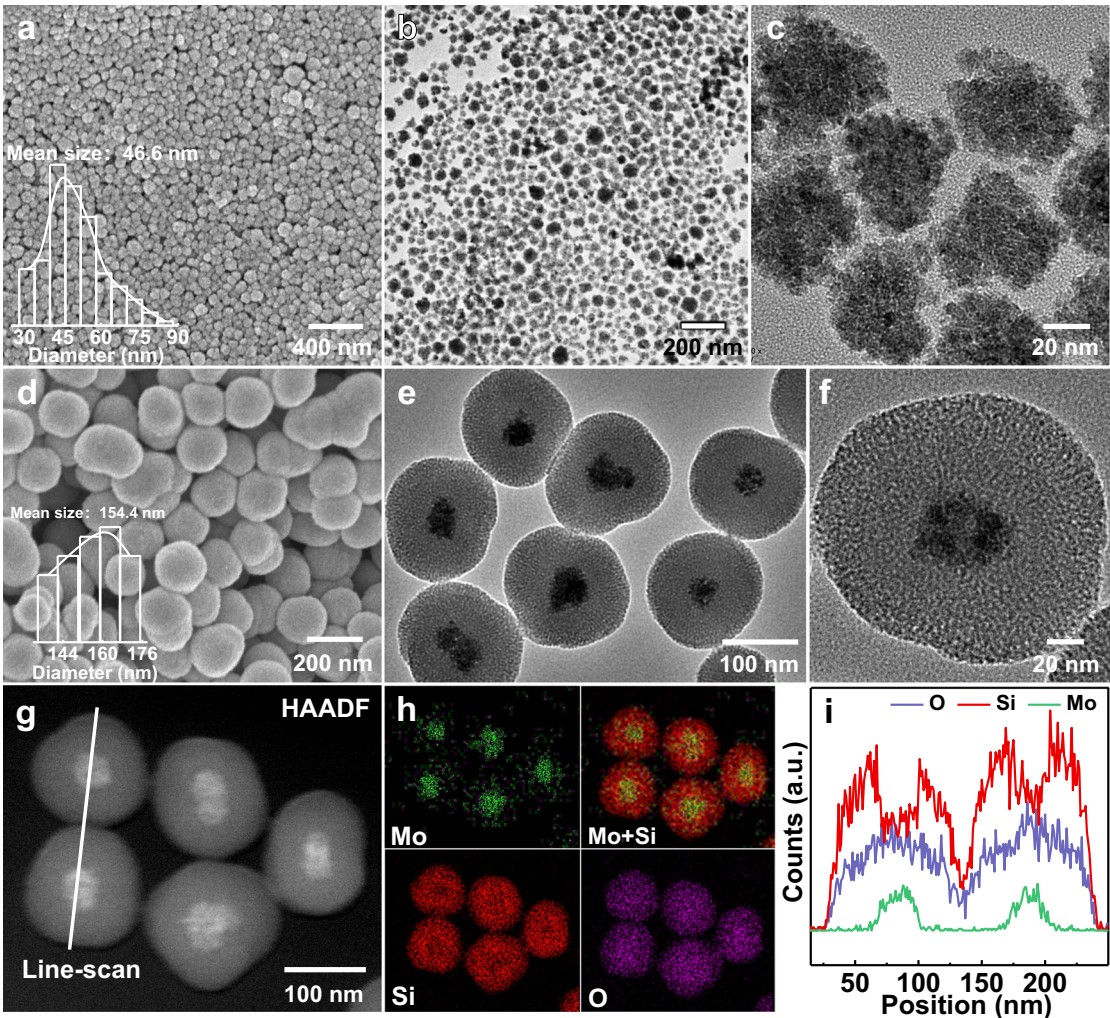

**Fig. 2 | Microscopy analysis of MoO$_2$ and MoO$_2$@SiO$_2$. a, b** FESEM (**a**) and TEM images (**b**–**c**) of MoO$_2$ nanocores. **d**–**i** FESEM (**d**), TEM images (**e, f**), HAADF-STEM image (**g**), corresponding elemental mappings (**h**), and EDX line profiles (**i**) of MoO$_2$@SiO$_2$. The EDX line profiles were along the white line of image (**g**).

conversion of MoO$_2$ cores to MoS$_2$ nanoparticles (MoS$_2$-NPs) without the intermediate silica encapsulation step results in the formation of thickly stacked MoS$_2$ sheets (~7 layers) extending outward from the original MoO$_2$ core (Supplementary Fig. 9). However, due to the absence of a silica shell to confine their growth, these sheets exhibit reduced curvature and presumably fewer defects compared to those formed within the MoS$_2$@SiO$_2$ catalysts. Additionally, the sheet stacks of MoS$_2$-NPs are prone to agglomeration, as particles with a size of 100–150 nm, approximately three times larger than the MoS$_2$ cores (40 nm) present in MoS$_2$@SiO$_2$ catalysts. Even thicker stacks (~10 layers) of MoS$_2$ can also be prepared by conventional hydrothermal method using AMT as Mo source and TAA as sulfur source (MoS$_2$-HT; Supplementary Fig. 10). The commercial MoS$_2$ powder (named MoS$_2$-Com) comprises aggregated MoS$_2$ crystals ranging from 300–400 nm (Supplementary Fig. 11), wherein only minimal quantities of stacked sheet morphology are observable. It is noteworthy to mention that the edges and in-plane sites of the MoS$_2$-NPs, MoS$_2$-HT, and MoS$_2$-Com are all less strained and much less accessible, so they cannot selectively expose in-plane Sv, which is unfavorable for methanol production.

Influences of MoO$_2$ diameter and SiO$_2$ shell thickness on the final morphology of MoS$_2$@SiO$_2$ were also studied systematically (Supplementary Fig. 12). By simply changing the amount of initially added AMT before hydrothermal synthesis of MoO$_2$ and changing the deposition reaction time of silica shell, average diameter of MoO$_2$ and shell thickness of SiO$_2$ can be precisely controlled. Firstly, MoO$_2$ core with

variable diameter (from 31 to 147 nm) can be encapsulated within silica shell (Supplementary Fig. 12a1–e2). Nevertheless, final morphology of MoS$_2$@SiO$_2$ can be altered after sulfidation. Only MoO$_2$ with a diameter between 31 nm and 66 nm can be successfully transformed into MoS$_2$@SiO$_2$ with fullerene-like MoS$_2$ (Supplementary Fig. 12a3–c3). Interestingly, the diameter of the generated MoS$_2$ is correlated positively with that of the pristine MoO$_2$ core. For MoO$_2$ with larger diameters, only the outside part of the MoO$_2$ core can be sulfidized to MoS$_2$ and the internal part is difficult to be completely transformed although the added TAA is in excess (Supplementary Fig. 12d3–e3). Subsequently, by extending the sulfidation time to 36 h, we find that the larger diameter MoO$_2$ (66 nm) can be fully sulfurized (Supplementary Fig. 13). HRTEM shows the formed MoS$_2$ is multilayered structure(6 - 8 layers) with strain. On the other hand, when the thickness of the silica shell is reduced to a certain extent, MoS$_2$ growth is observed on the external surface of the silica shell under the same synthetic conditions (Supplementary Fig. 14a3–c3). Therefore, we conclude that the final morphology of MoS$_2$@SiO$_2$ depends heavily on the above two synthetic parameters. Thus, it is important to strike a balance for both size of MoO$_2$ core and thickness of silica shell to obtain the confined few-layer MoS$_2$ hollow sphere. Moreover, we also find that the SiO$_2$ shell of MoS$_2$@SiO$_2$ can be easily removed with KOH solution. HRTEM image of obtained MoS$_2$ (MoS$_2$-R) shows no fullerene-like structure, but rather randomly aggregated and stacked structure (Supplementary Fig. 15). This means that the SiO$_2$ shell not

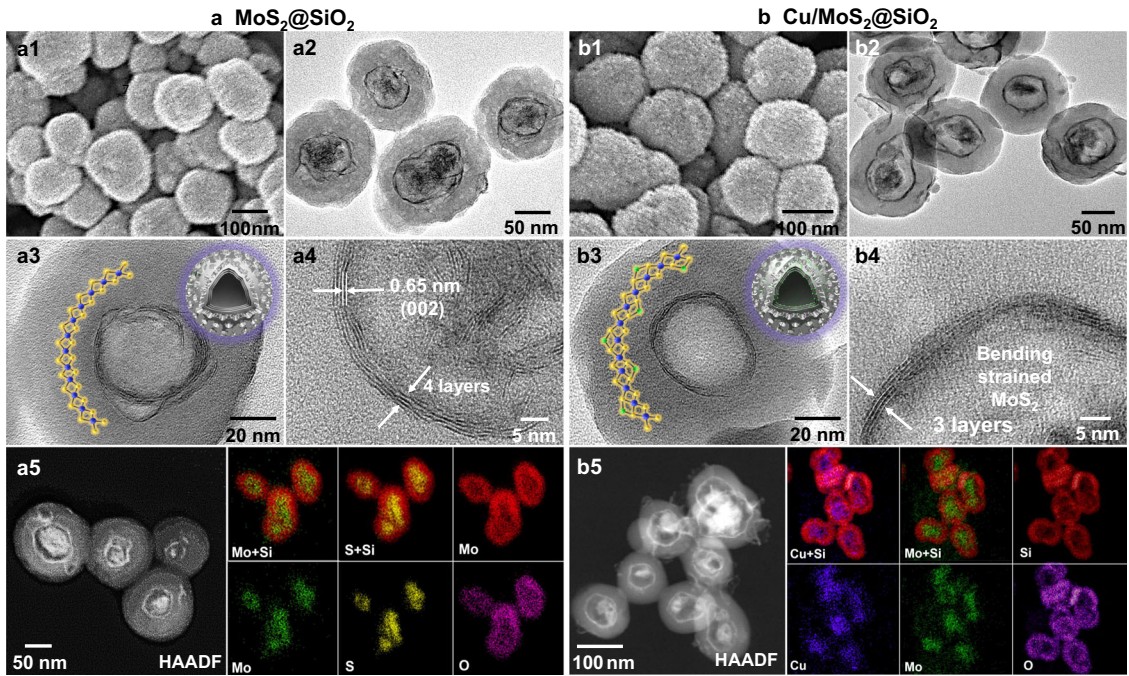

**Fig. 3 | Microscopy analysis of MoS$_2$@SiO$_2$ and Cu/MoS$_2$@SiO$_2$. a1**–**a5**, FESEM (**a1**), TEM images (**a2**–**a4**), HAADF-STEM image and elemental mappings (**a5**) of MoS$_2$@SiO$_2$. **b1**–**b5**, FESEM (**b1**), TEM images (**b2**–**b4**), HAADF-STEM image and elemental mappings (**b5**) of Cu/MoS$_2$@SiO$_2$.

only confines the transformation of MoO$_2$ cores to MoS$_2$ hollow spheres, but also isolates MoS$_2$ to prevent it from aggregating and maintains its strained, few-layer and fullerene-like structure.

As we mentioned earlier, the strain in fullerene-like MoS$_2$ is induced by its surface curvature, which in turn is contingent upon the average particle size of MoO$_2$. To illuminate the correlation between curvature and strain, the in-plane uniaxial strain ($S$) of the prepared MoS$_2$@SiO$_2$ with spherical MoS$_2$ radius from 15.9 nm to 31 nm is calculated based on geometric analysis in Supplementary Fig. 16. For example, the formula for a preliminary estimation of interlayer MoS$_2$ strain is defined as follows: $S = -d/r$, where $r$ is the radius from the hollow sphere origin to a strained MoS$_2$ layer and $d$ is the interlayer distance between two MoS$_2$ layers under comparison. Based on the number of MoS$_2$ layers or the thickness of MoS$_2$ and the radius of the spherical MoS$_2$, the uniaxial strain of MoS$_2$ (radius ($r$) of 15.9 nm, 22.4 nm, and 31 nm) in prepared MoS$_2$@SiO$_2$ are calculated to be $S = -4.1\%$, $-2.9\%$ and $-2.1\%$ (compressive strains), respectively (Supplementary Fig. 16c). This calculation is only for a simple and visual comparison of the compressive strain because of more complex biaxial strains in spherical fullerene-like MoS$_2$ in MoS$_2$@SiO$_2$, which will be further addressed in our DFT calculations later for intralayer strain analysis of MoS$_2$. Based on this illustration, nevertheless, we can conclude the strain of MoS$_2$@SiO$_2$ can be simply and easily adjusted by changing the particle size of initial MoO$_2$ nanocores.

The crystallographic structures of different samples were determined via powder X-ray diffraction (XRD) analysis (Fig. 4a, Supplementary Fig. 17). The XRD peaks observed for pristine MoO$_2$ nanocores at 36.5, 41.45, 53.7, and 65.7° correspond to the monoclinic phase of MoO$_2$ (JCPDS 50-0739). The observed reduction in peak intensity in MoO$_2$@SiO$_2$ can be attributed to the lower MoO$_2$ core content within the SiO$_2$ shell. For MoS$_2$@SiO$_2$ and MoS$_2$-NPs, two apparent broad peaks at 32.3 and 57.0° are assigned to (100) and (110) reflections of MoS$_2$ in 2H polymorph (JCPDS 37-1492). Notably, the MoS$_2$@SiO$_2$ sample does not exhibit the (002) peak at 14.5°, which typically correspond to the periodicity in $c$-axis direction (normal to the MoS$_2$ basal plane). The presence of (002) reflection is indicative of multilayer MoS$_2$ sheets, and it has been demonstrated to be absent in the case of single-layer or few-layer MoS$_2$. Based on this observation, in conjunction with the TEM analysis, we deduced that the spherical MoS$_2$ structures formed in MoS$_2$@SiO$_2$ are indeed few-layered, whereas those in MoS$_2$-NPs are multilayered. On the contrary, XRD patterns of commercial MoS$_2$ and MoS$_2$-HT (Supplementary Fig. 17) exhibit significantly greater intensity in the (002) reflection compared to other peaks, signifying a high degree of stacking for the MoS$_2$ sheets in these two samples. In addition, the peak of Cu/MoS$_2$@SiO$_2$ is basically the same as MoS$_2$@SiO$_2$. This is due to the presence of relatively low Cu content and the effective dispersion of Cu species in MoS$_2$@SiO$_2$. Furthermore, N$_2$ physisorption analysis (Fig. 4b) elucidates that Cu/MoS$_2$@SiO$_2$ has a type IV physisorption isotherm with a type H4 hysteresis loop, a characteristic feature of mesoporous silica. This sample displays a BET surface area of 98.1 m$^2$/g, a pore volume of 0.15 cm$^3$/g, and a pore-size range from 2.0 to ~18 nm, thus affirming the existence of mesopores.

Next, X-ray photoelectron spectroscopy (XPS) was employed to investigate the chemical state and electronic structure of our samples. In Supplementary Fig. 18a, the doublets located at 229.6 and 232.7 eV are assigned to Mo 3$d_{3/2}$ and Mo 3$d_{5/2}$ components, respectively, while the shorter peak at 226.7 eV is attributed to S 2$s$ shell electrons. The presence of weak Mo$^{6+}$ components is attributable to slight oxidation of the samples in air. Additionally, the high-resolution S 2$p$ XPS spectra of all samples (Supplementary Fig. 18b) display well-resolved S 2$p_{3/2}$ and S 2$p_{1/2}$ doublets, with peak positions at 162.3 eV and 163.5 eV, respectively. Compared with MoS$_2$-Com, MoS$_2$-HT, and MoS$_2$-NPs, consistently, MoS$_2$@SiO$_2$ and Cu/MoS$_2$@SiO$_2$ samples show much lower signal intensities of Mo 3$d$ and S 2$p$ photoelectrons, which confirm that almost all their hollow cores of MoS$_2$ are confined within the mesoporous shell of SiO$_2$ in the latter two samples.

Furthermore, electron paramagnetic resonance (EPR) measurements were conducted to analyze and compare the Sv number in the various catalysts (Fig. 4c). The EPR technique allows for the detection of paramagnetic signals and the signal observed at approximately 330 mT ($g = 2.0$) provides information about the concentration of unsaturated sites with unpaired electrons, which is directly proportional to the content of Sv in the tested catalysts[43,44]. Clearly,

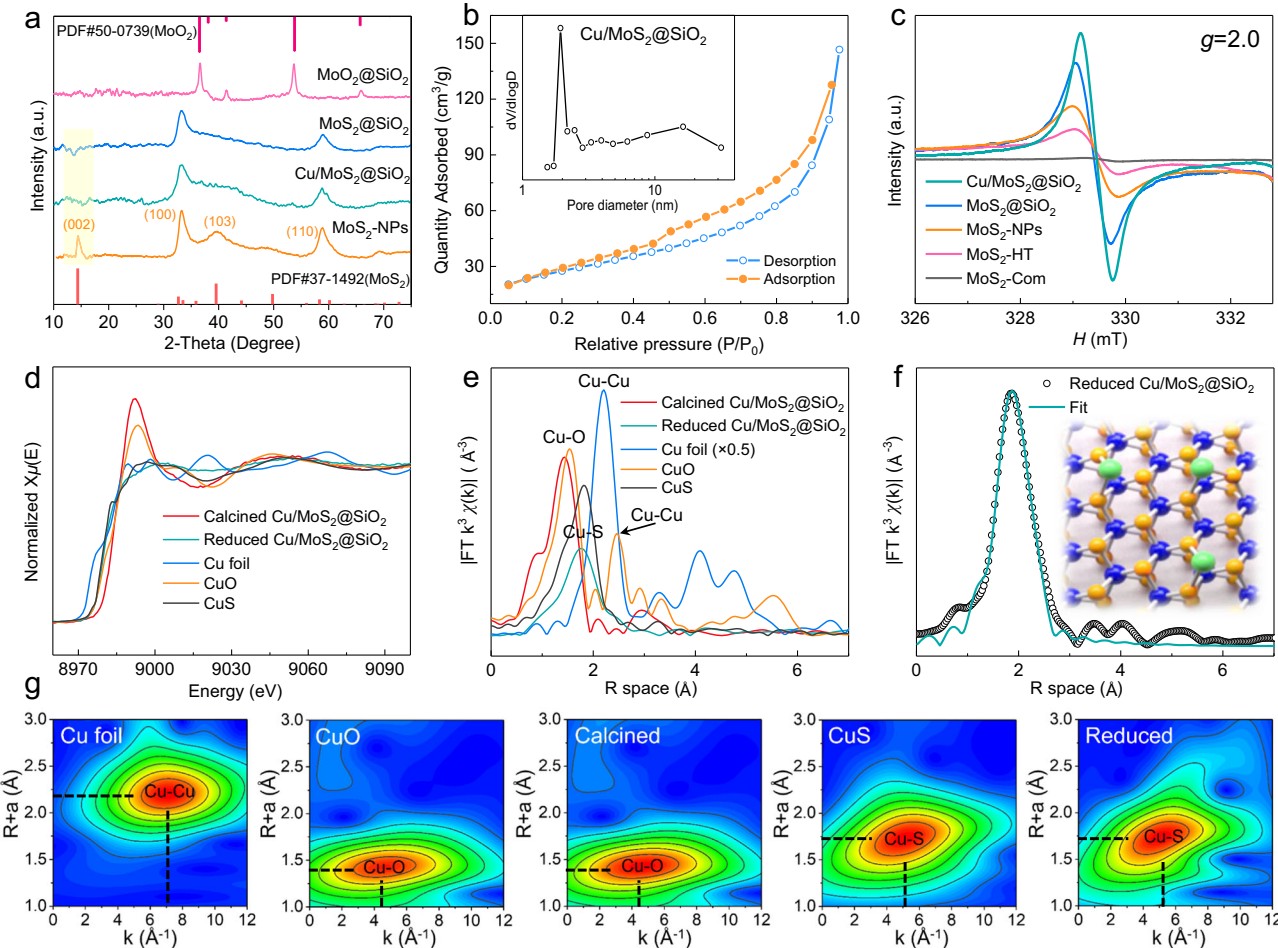

**Fig. 4 | Characterization of different catalysts. a–c** XRD patterns (**a**), Nitrogen sorption isotherms, and the corresponding pore-size distribution (**b**) and EPR spectra of different samples (**c**). **d**, **e** Cu K-edge normalized XANES spectra (**d**) and FT $k^3$-weighted Cu K-edge EXAFS spectra (**e**) of calcined and reduced Cu/ MoS$_2$@SiO$_2$ and the references. **f** Corresponding FT-EXAFS fitting curves of reduced Cu/MoS$_2$@SiO$_2$ in $R$ space. **g** WT-EXAFS plots of calcined and reduced Cu/ MoS$_2$@SiO$_2$ and the references.

MoS$_2$@SiO$_2$ displays the significantly higher peak intensity than MoS$_2$-NPs, MoS$_2$-HT, and MoS$_2$-Com, implying a more pronounced charge-compensating effect and consequently a higher concentration of Sv in MoS$_2$@SiO$_2$. Therefore, few-layer fullerene-like MoS$_2$ with higher strains favors the generation of more Sv. In addition, the peak area and intensity of Cu/MoS$_2$@SiO$_2$ are noticeably higher than that of MoS$_2$@SiO$_2$, indicating the formation of even more Sv in proximal sites after the introduction of Cu[43].

Raman spectroscopy analysis was also employed to investigate the structural characteristics of the prepared catalysts. As seen in Supplementary Fig. 19, all samples exhibit three prominent MoS$_2$ Raman shifts at 383.0, 407.7, and 455.2 cm$^{-1}$, which correspond to the in-plane Mo-S phonon mode (E$_{2g}^1$), the out-of-plane Mo-S mode (A$_{1g}$), and the second-order Raman scattering 2LA(M), respectively[45]. The positions of A$_{1g}$ peaks of MoS$_2$@SiO$_2$ and Cu/MoS$_2$@SiO$_2$ are found to be almost identical. This indicates that Cu is not located in the lattice of MoS$_2$. Because the substitutional replacement of Mo sites in basal planes by other metal atoms could soften the Mo-S modes and lead to a decrease in their vibration frequency, causing red shifts of E$_{2g}^1$ and A$_{1g}$ peaks[45,46].

To reveal the electronic structure and coordination environment of Cu species in Cu/MoS$_2$@SiO$_2$ at atomic level, synchrotron-radiation-based X-ray absorption spectra (XAS) were further measured, encompassing X-ray absorption near-edge structure (XANES) and the extended X-ray absorption fine structure (EXAFS). The ex situ

measured XAS spectra of standard samples (Cu foil, CuS, and CuO) were also provided for comparison. The Cu K-edge XANES spectra of calcined Cu/MoS$_2$@SiO$_2$ virtually obey that of CuO reference profile (Fig. 4d), revealing the formation of copper oxide after calcination. Quite interestingly, the energy spectra of Cu/MoS$_2$@SiO$_2$ sample after reduction is higher than that of Cu foil, hence demonstrating the average valence state of Cu being equal +2. In the Fourier transform EXAFS (FT-EXAFS) spectra of the reduced sample (Fig. 4e), a prominent single strong shell at ca. 1.77 Å in $R$-space is found, implying the formation of Cu−S bond. When compared with Cu foil, the typical Cu−Cu peak is not detected in the spectra, suggesting the atomic dispersion of Cu species in Cu/MoS$_2$@SiO$_2$. According to the EXAFS fitting curves (Fig. 4f) and fitting parameters (Supplementary Fig. 20, Supplementary Table 2), the coordination number (CN) of Cu−S bond is calculated to be 1.5, meaning one copper atom is bonded to one or two sulfur atoms with a respective bond length of 2.22 Å. Since the Mo CN of standard MoS$_2$ equals 6 and Mo CN of Sv-rich MoS$_2$ equals around 5, the Cu CN of 1.5 indicate that Mo atoms are not replaced with atomic Cu[25]. Therefore, it can be inferred that the surface-loaded Cu atoms form partial bonding with near-surface sulfur species present in MoS$_2$ crystal structure upon hydrogen reduction at elevated temperature (insert of Fig. 4f). Moreover, it is worth noting that small amounts of Cu nanoparticles/clusters are still present in the samples based on the linear combination fitting (Supplementary Table 3). Besides, the wavelet transforms (WT) analysis of Cu K-edge EXAFS

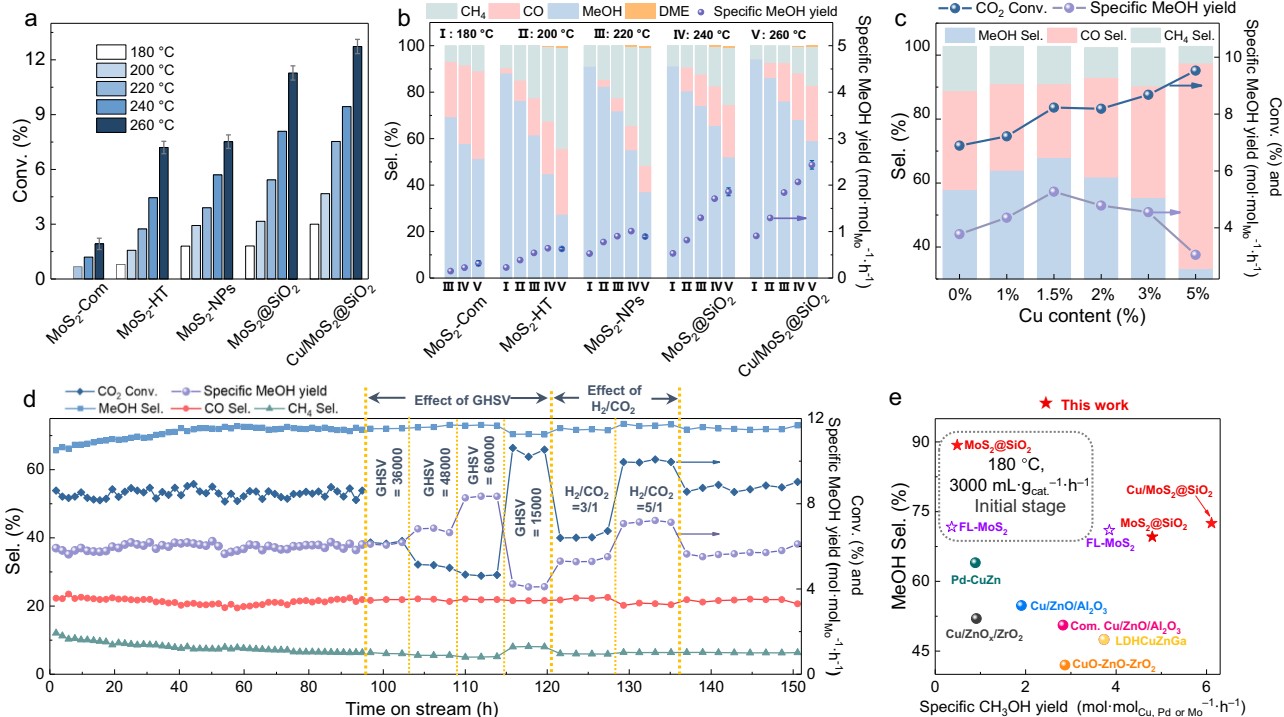

**Fig. 5 | Catalytic performance comparison of different catalysts in CO₂ hydrogenation to methanol. a, b** Conversion (**a**), selectivity, and specific MeOH yield (**b**) for CO₂ hydrogenation of various MoS₂-based catalysts at different reaction temperatures (180–260 °C). Reaction conditions: 5 MPa, H₂:CO₂ = 4:1, GHSV = 8000 mL g$_{cat.}$$^{-1}$ h$^{-1}$. The data at 260 °C in **a–b** were collected three times, and the error bars represent the standard deviation. **c** Catalytic performance comparison of Cu/MoS₂@SiO₂ catalysts with different Cu content. Reaction conditions: 5 MPa, H₂:CO₂ = 4:1, GHSV = 24000 mL g$_{cat.}$$^{-1}$ h$^{-1}$. **d** Long-term test of Cu/MoS₂@SiO₂. Standard reaction conditions: 260 °C, 5 MPa, H₂:CO₂ = 4:1, GHSV = 24000 mL g$_{cat.}$$^{-1}$ h$^{-1}$. **e** Comparison of the MeOH selectivity and specific MeOH yield over Cu/MoS₂@SiO₂, MoS₂@SiO₂, and other state-of-the-art catalysts under similar reaction conditions (see Supplementary Table 5 for more details).

oscillations and K-edge EXAFS k²χ functions is performed to visually verify the above findings (Fig. 4g). As compared with the Cu foil, CuO, and CuS references, the maximum of WT contour plots at ~4.1 Å$^{-1}$ is assigned to the Cu–O bonds in calcined sample, whereas the maximum of WT contour plots at ~4.6 Å$^{-1}$ correspond to Cu–S bonds in the reduced sample, which is consistent with the EXAFS results. All these observations confirm the presence of atomic Cu over Cu/MoS₂@SiO₂.

## Comparison of catalytic performance

All the studied samples (MoS₂-Com, MoS₂-HT, MoS₂-NPs, MoS₂@SiO₂ and Cu/MoS₂@SiO₂) were evaluated for CO₂ hydrogenation to methanol within a temperature range of 180–260 °C and at a gaseous hourly space velocity (GHSV) of 8000 mL g$_{cat.}$$^{-1}$ h$^{-1}$. Throughout the reaction system, methanol served as a primary product, while CO and CH₄, with a trace amount of dimethyl ether (DME), were detected as by-products in all experiments. Figure 5a, b report CO₂ conversion, methanol selectivity, and the specific MeOH yield on these samples. As anticipated, the CO₂ conversion rises with increasing reaction temperature while the methanol selectivity declines across these samples. This trend can be attributed to the exothermic nature of the methanol production reaction (CO₂ + H₂ → CH₃OH + H₂O, $\Delta\hat{H}_{298\ K} = -49.5$ kJ mol$^{-1}$), which becomes thermodynamically unfavorable at higher reaction temperatures. Besides, the observed increase in CO selectivity with elevated reaction temperature can be ascribed to the occurrence of the reverse water-gas shift reaction (RWGS, CO₂ + H₂ → CO + H₂O, $\Delta\hat{H}_{298\ K} = 41.2$ kJ mol$^{-1}$), a significant parallel reaction during CO₂ hydrogenation that becomes thermodynamically favorable at higher temperatures. Among these catalysts, bulk MoS₂-Com only provides a CO₂ conversion of 1.92% with a methanol selectivity of 51.3% at 260 °C. Both MoS₂-HT and MoS₂-NPs also give unsatisfactory

conversion (<8%) and methanol selectivity (<40%) with a specific MeOH yield of 0.64 and 0.89 mol$_{MeOH}$ mol$_{Mo}$$^{-1}$ h$^{-1}$, respectively at 260 °C. Gratifyingly, the as-fabricated MoS₂@SiO₂ show a much better CO₂ conversion and methanol selectivity than MoS₂-Com, MoS₂-HT, and MoS₂-NPs over 180–260 °C. Besides, MoS₂@SiO₂ also exhibit significantly lower methane selectivity than the three other catalysts. Under our optimal reaction conditions, MoS₂@SiO₂ can exhibit a specific MeOH yield up to 1.89 mol$_{MeOH}$ mol$_{Mo}$$^{-1}$ h$^{-1}$ with CO₂ conversion of 11.28% and methanol selectivity of 52.16% at 260 °C and GHSV of 8000 mL g$_{cat.}$$^{-1}$ h$^{-1}$. The effect of molecular MoS₂ sheet strain on performance is further examined and the corresponding findings are displayed in Supplementary Fig. 21. It can be clearly seen that the MoS₂@SiO₂ with higher strain obtain better CO₂ conversion along with similar methanol selectivity, which can be attributed to the fact that the formation of Sv will be easier for the catalysts under higher strain. Considering the lower yield during the preparation of MoO₂ with small particle size of 47.6 nm, hereafter we used MoS₂@SiO₂ with strain of −2.9% for more experiments. We also tested the catalytic performance of MoS₂-R sample (the silica shell was removed), which showed significantly lower conversion and methanol selectivity than MoS₂@SiO₂ (Supplementary Table 4). This result further confirms the importance of strained fullerene-like structure of MoS₂ with selectively exposed in-plane Sv for efficient methanol synthesis.

Metal promotion is a pivotal approach in CO₂ hydrogenation to methanol, aimed at enhancing activity by fostering H₂ activation[24,47]. When 1.5 wt% Cu is further introduced into MoS₂@SiO₂ catalyst, both CO₂ conversion and methanol selectivity are improved. This Cu/MoS₂@SiO₂ catalyst displays a higher CO₂ conversion of 12.73%, methanol selectivity of 59.2%, and specific MeOH yield of 2.42 mol$_{MeOH}$ mol$_{Mo}$$^{-1}$ h$^{-1}$, suggesting that Cu species is favorable for promoting the methanol synthesis from CO₂ hydrogenation. We have

also tested the performance of the reference catalyst Cu@SiO₂ (Supplementary Table 4). The result shows a low activity of Cu@SiO₂, with only 2.5% $CO_2$ conversion and 50.4% methanol selectivity. This suggests that the $Cu/MoS_2$ interface is essential for driving methanol synthesis. Subsequently, we further investigated the effect of Cu loading on the catalytic performance of $CO_2$ hydrogenation at GHSV of 24,000 mL $g_{cat.}^{-1}$ $h^{-1}$. As depicted in Fig. 5c, it is found that $CO_2$ conversion increases monotonously with the increment of Cu content, while the selectivity of methanol and specific MeOH yield exhibit a volcano curve pattern in response to the varying Cu content. On the other hand, a gradual increase of Cu content to greater than 2 wt% also boosts the RWGS reaction and hence the CO selectivity can be increased. The optimized $1.5\%Cu/MoS_2@SiO_2$ catalysts display the highest methanol selectivity (66.6%) and specific MeOH yield (5.30 $mol_{MeOH}$ $mol_{Mo}^{-1}$ $h^{-1}$) at GHSV of 24000 mL $g_{cat.}^{-1}$ $h^{-1}$. As for catalytic $CO_2$ hydrogenation to methanol over $MoS_2$-based catalysts, there are several factors that significantly affect the performance of this catalyst: (1) fewer layer number of $MoS_2$ sheets and higher density of Sv; (2) more exposed in-plane Sv and less exposed edge Sv; and (3) the introduction of transition metal in $MoS_2$ enhances $H_2$ activation and spillover, which can promote $CO_2$ conversion[24,48–52]. In our catalytic system, the superior performance of $1.5\%Cu/MoS_2@SiO_2$ can be attributed to three primary factors, as supported by the characterization results: (1) benefitting from the as-synthesized ultrafine $MoO_2$ nanocores and the silica-encapsulated core-shell configuration, the generated $MoS_2$ has a buckyball structure with only 2 to 4 layers, which significantly contributes to the generation and exposure of high-density Sv; (2) physically constrained topologic growth of $MoS_2$ nanosheets within the central cavity of spherical mesoporous silica promotes the generation of curvature-induced in-plane strain and Sv into the original inert $MoS_2$ basal plane; and (3) fullerene-like few-layer $MoS_2$ hollow sphere can selectively expose more in-plane Sv, simultaneously reducing the exposure of edge Sv. Thus, this unique $MoS_2$ structure is extremely favorable for the selective synthesis of methanol; and (4) the bending behavior of $MoS_2$ can also introduce sufficient accessible sites to anchor Cu. For example, the moiety of sulfur-chelated Cu may act as an active site in hydrogen spillover process and the presence of Cu−S may also facilitate the formation of in-plane Sv and thus lead to catalytic synergy of Cu-$MoS_2$ in methanol synthesis.

Long-term stability tests of $MoS_2@SiO_2$ and $Cu/MoS_2@SiO_2$ were also carried out for a total duration of 150 h at 260 °C (Fig. 5d, Supplementary Fig. 22). The $CO_2$ conversion and methanol selectivity over both $MoS_2@SiO_2$ and $Cu/MoS_2@SiO_2$ catalysts increase relatively fast in the initial stage of the reaction process, followed by a gradual rise, suggesting that the reductive reaction gas facilitates the formation of additional Sv. Furthermore, the EPR method was employed to characterize the Sv of the recovered catalysts (Supplementary Fig. 23a). The results indicate that the used catalysts exhibit markedly stronger signal intensity compared to the $H_2$-pretreated fresh catalysts, providing further validation for the previous deduction. Over the extended reaction time of 150 hours, both $MoS_2@SiO_2$ and $Cu/MoS_2@SiO_2$ demonstrate remarkable stability with no discernible decline in $CO_2$ conversion, methanol selectivity, and specific MeOH yield, indicating the great potential for practical application. Subsequent investigation was performed on the recovered $MoS_2@SiO_2$ and $Cu/MoS_2@SiO_2$ catalysts after the extended-duration experiments. Our HRTEM results (Supplementary Figs. 24–25) confirm the preservation of the hollow silica-encapsulated core-shell structure and few-layer fullerene-like $MoS_2$ in both $MoS_2@SiO_2$ and $Cu/MoS_2@SiO_2$ catalysts after a 150-hour testing period. The XRD patterns and XPS spectra of the used catalysts (Supplementary Fig. 23b-d) show that the diffraction peaks of the $MoS_2$ phase are almost identical to those of the fresh ones. It is thus evident that the structure and composition of $MoS_2@SiO_2$ and $Cu/MoS_2@SiO_2$ are indeed robust. During the stability evaluation of $Cu/MoS_2@SiO_2$, the effect of GHSV and $H_2:CO_2$ ratio was examined at

260 °C. Rather expected, we observed that the $CO_2$ conversion of this sample decreases with the increase in GHSV, whereas its methanol selectivity and specific MeOH yield increase. This phenomenon can be attributed to the limitation of the reverse water-gas shift (RWGS) reaction at a high GHSV, where a short contact time can suppress the occurrence of the RWGS reaction. Moreover, we found that the decrease of $H_2:CO_2$ ratio to 3:1 would result in a slight decrease in $CO_2$ conversion and specific MeOH yield, and higher $H_2:CO_2$ ratio (5/1) was beneficial to the $CO_2$ conversion and methanol selectivity. We also evaluated our $Cu/MoS_2@SiO_2$ catalyst under the similar reaction conditions with other state-of-the-art catalysts. Generally, the induction period of oxide catalysts in $CO_2$ hydrogenation was very short. However, for $MoS_2$ catalysts, a prolonged induction period was observed during the initial stage of the $CO_2$ hydrogenation process[25]. During the initial 400 h of the stability test, both $CO_2$ conversion and methanol selectivity kept increasing slowly[25]. Therefore, for a fair comparison, we compared the performance of the prepared $MoS_2@SiO_2$ catalysts with FL-$MoS_2$ at the early stage of the reaction after 3 h of $H_2$ reduction at 180 °C. As shown in Fig. 5e and Supplementary Table 5, $MoS_2@SiO_2$ show a significantly higher methanol selectivity at the early stage of the reaction. This further confirms that the selective exposure of in-plane Sv of $MoS_2@SiO_2$ plays a crucial role in the high selective methanol synthesis from $CO_2$ hydrogenation. As far as our knowledge extends, both specific MeOH yield (6.11 mol $mol_{Mo}^{-1}$ $h^{-1}$) and methanol selectivity (72.5%) of our $Cu/MoS_2@SiO_2$ under a GHSV of 24,000 mL $g_{cat.}^{-1}$ $h^{-1}$ at 260 °C after 150 h on stream in this study are notably higher than those of other recently reported Mo, Cu-based catalysts including commercial $Cu/ZnO/Al_2O_3$ during $CO_2$ hydrogenation. Considering the fact that with the extremely long on-stream test time, the activity of $Cu/MoS_2@SiO_2$ and $MoS_2@SiO_2$ catalysts was still slowly increasing, we inferred that the final conversion and selectivity would be higher after 3000 h based on the conclusion of previous literature[25]. On the other hand, the catalytic performance of the prepared catalysts was evaluated at lower reaction temperatures and pressures (230 °C, 25 bar; Supplementary Fig. 26). The result shows that our prepared catalysts also exhibit higher methanol selectivity than others in the recent literature under mild conditions. This further confirms that the selective exposure of in-plane Sv of $Cu/MoS_2@SiO_2$ play a crucial role in highly selective methanol synthesis from $CO_2$ hydrogenation.

## The role of Cu and in situ spectroscopic evidence for reaction mechanism

Based on our previous EPR spectra, we concluded that the introduction of Cu can promote the formation of Sv. The effect of Cu on Sv formation on $MoS_2$ basal plane was further investigated using DFT. We have identified that Mo atop and hollow sites are the most thermodynamically stable sites for Cu single atoms on $MoS_2$ basal plane (Supplementary Table 6). The proximity to Cu dopants decreases Sv formation energies by 1.5 eV (Fig. 6a, b). Notably, S atoms without direct coordination with Cu also exhibit lower vacancy formation energies (Fig. 6a, b). Charge density difference plots indicate a depletion of electron in the region of Mo-S bonds of S atoms coordinated to Cu, which weakens the Mo-S bonds and promotes S removal (Supplementary Fig. 27). Following the removal of a neighboring S atom, Cu fills the resulting vacancy upon the geometry optimization. The formation of the second Sv in this structure is also easier than in pristine $MoS_2$, especially, next to the Cu dopant (Supplementary Fig. 28). The Cu atom exhibits facile migration between these two Sv sites, with a low Gibbs energy barrier of 0.33 eV (Fig. 6c). In the corresponding transition state, Cu is coordinated to one S atom with a Cu−S bond length of 2.29 Å.

Subsequently, we conducted high-pressure in situ DRIFTS tests using our best catalysts to gain insights into major intermediates and consequently to understand the reaction mechanism of $CO_2$

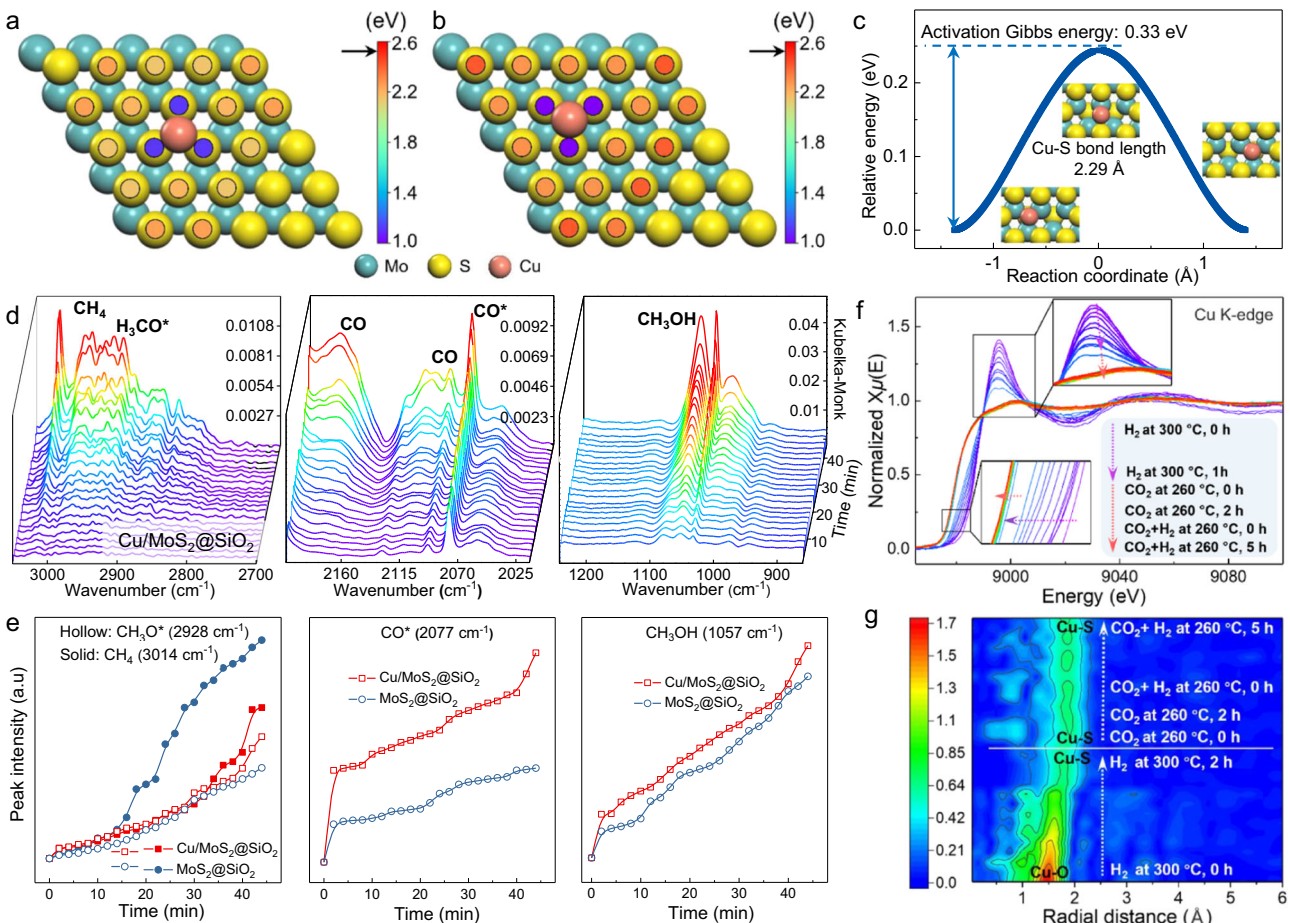

**Fig. 6 | DFT studies on the role of Cu and in situ characterizations for reaction mechanism. a**, **b** Sv formation energies of different S sites when a Cu atom is anchored at (**a**) Mo atop, and (**b**) hollow sites. The arrow indicates the energy value in the absence of Cu. **c** Cu migration within the two-vacancy site. Inset: atomic structures of the initial, transition, and final states. **d** High-pressure in situ DRIFT spectra of the $CO_2$ hydrogenation to methanol reaction catalyzed by Cu/$MoS_2$@$SiO_2$; reaction conditions: 20 mg catalyst, 30 bar, 250 °C, 20 mL min$^{-1}$

reaction gas ($H_2$:$CO_2$ = 3:1). **e** Corresponding IR peak intensities of various species versus time during in situ $CO_2$ + $H_2$ reaction. **f**, **g** *Operando* XAS spectra of Cu K-edge Cu/$MoS_2$@$SiO_2$ under high-pressure $CO_2$ hydrogenation to methanol. XANES spectra (**f**) and continuous contour-plot of corresponding $k^2$-weighted R-space EXAFS spectra (**g**); reaction conditions: 10 bar, 260 °C, $H_2$:$CO_2$ = 3:1. The arrows in **f**, **g** help to denote the variations of reaction condition and gas composition along the test process.

hydrogenation. Figure 6d and Supplementary Fig. 29 show the collection of DRIFTS spectra during the hydrogenation of $CO_2$ at 250 °C and 30 bar using Cu/$MoS_2$@$SiO_2$ and $MoS_2$@$SiO_2$ catalysts. Both catalysts give very similar absorbance bands, suggesting the same reaction intermediates and similar transformation pathways of $CO_2$ to methanol. Hence, the benefitting role of Cu is only as a promoter. Upon the introduction of the feed gas mixture, the DRIFTS spectra exhibit the presence of linearly bridged CO* species at approximately 2076 cm$^{-1}$, Mo=O species with peaks ranging from 900 to 965 cm$^{-1}$, and Mo−O−Mo species with a broad band observed at around 700 to 865 cm$^{-1}$ in the temperature range of 180 to 260 °C. These observations suggest that the coordinatively unsaturated Mo sites play an effective role in dissociating $CO_2$ into *CO and *O species on highly active Sv, as illustrated in Supplementary Fig. 30[25,53–55]. The vibrational bands at 2175 and 2115 cm$^{-1}$ are identified as gaseous CO, signifying the occurrence of the reverse water-gas shift reaction (RWGS)[55,56]. Moreover, the characteristic peaks in the range of 2810 to 3000 cm$^{-1}$ can be assigned to $\nu$($CH_3$) modes of $CH_3O$* species[25,56]. Furthermore, bands at 1054, 1033, and 1005 cm$^{-1}$, which are associated with the C−O stretch of methanol, can also be observed, providing evidence of methanol generation[57]. The absorbance band observed at around 3014 cm$^{-1}$ is assigned to methane. The bands at 3014, 2928, 2076, and 1054 cm$^{-1}$ were selected to study the time-dependent variations of $CH_4$, $CH_3O$*, CO*, and $CH_3OH$ at 240 °C. The intensities of these peaks exhibit an

increment with the extension of test time, indicating the progressive formation and accumulation of intermediates during the prolonged reaction (Fig. 6e). For comparative purposes, in situ DRIFTS tests were also performed with $MoS_2$-NPs without $SiO_2$ shell (Supplementary Fig. 31). Our result confirms that this reference catalyst also has the similar bands, suggesting the same hydrogenation reaction mechanisms over it. However, the intensity of all the peaks of $MoS_2$-NPs is significantly weaker than those of Cu/$MoS_2$@$SiO_2$ and $MoS_2$@$SiO_2$, which further demonstrates the inferior catalytic activity of $MoS_2$-NPs. Interestingly, compared with $MoS_2$@$SiO_2$, the intensity of methane characteristic peak in Cu/$MoS_2$@$SiO_2$ catalysts is weaker, which further proved that the addition of Cu could promote the formation of methanol and suppress the excessive hydrogenation of $CO_2$ to $CH_4$. Furthermore, the IR band intensities of $CH_3O$*, CO*, and $CH_3OH$ over Cu/$MoS_2$@$SiO_2$ are much greater than those of $MoS_2$@$SiO_2$. This observation suggests that the presence of Cu enhances the formation of CO* from chemisorbed $CO_2$ and facilitates its subsequent hydrogenation to $CH_3O$* and $CH_3OH$ (Fig. 6e). To gain further insights into the enhancing effect of Cu on catalytic performance, we carried out an investigation on the adsorption and desorption of $H_2$ on Cu/$MoS_2$@$SiO_2$ and $MoS_2$@$SiO_2$ catalysts. $H_2$ temperature-programmed desorption ($H_2$-TPD) experiments were performed, as they would offer valuable information regarding the adsorbed hydrogen species and their relative concentrations at the catalyst surface[58]. As depicted in

Supplementary Fig. 32, both catalysts exhibit a dominant peak around 300 °C, corresponding to the desorption of weakly adsorbed hydrogen species, likely attributed to weakly dissociatively adsorbed H species[49,59,60]. The higher desorption capacity of $H_2$ observed for Cu/$MoS_2$@$SiO_2$ in comparison to $MoS_2$@$SiO_2$ suggests a higher concentration of adsorbed H species on Cu/$MoS_2$@$SiO_2$. Hence, the promoting effect of Cu can be primarily attributed to its facilitation of Sv generation, $H_2$ dissociation and activation, as supported by our findings from ESR spectra, in situ DRIFT, $H_2$-TPD, and DFT calculations.

Separately from dynamic intermediate formation revealed by using the in situ DRIFTS during methanol synthesis over the surface of Cu/$MoS_2$@$SiO_2$ catalyst, *operando* XANES and EXAFS experiments at the Cu K-edge were also performed to investigate the real electronic state, coordination environment and the dynamic evolution of Cu species as catalytically active sites under the $CO_2$ hydrogenation condition. Briefly, pre-calcined Cu/$MoS_2$@$SiO_2$ underwent three consecutive stages including: (i) reduction under $H_2$ at 300 °C, (ii) exposure to pure $CO_2$ at 260 °C, and (iii) exposure to reaction gas ($H_2$:$CO_2$ = 3:1, 10 bar) at 260 °C. The continually collected Cu K-edge XANES data of Cu/$MoS_2$@$SiO_2$ undergoing various stages show a gradually descending white line intensity and edge position under $H_2$ at 300 °C (Fig. 6f). After 2 h of reduction under hydrogen, the curve position and intensity stabilized and even remain almost unchanged by swapping $H_2$ to pure $CO_2$ and reaction gas whose flow lasted for 5 h. Notably and importantly, the corresponding contour-based plot of FT-EXAFS analysis (Fig. 6g) also show the continual transformation of Cu −O coordination (1.45 Å) to Cu−S coordination (1.57 Å), as experimental analysis proceed. These observations suggest that Cu atoms coordinated directly with the S species in $MoS_2$ of Cu/$MoS_2$@$SiO_2$ upon $H_2$ treatment. Indeed, no notable change of the Cu−S shells was found for a prolonged catalyst exposure to $CO_2$ and reaction gas, indicating that no metallic copper nanoparticles or copper oxide were formed, which in turn meant a high stability of atomically dispersed Cu (coordinated by S atoms) on the nanosheet surface of $MoS_2$ during the $CO_2$ hydrogenation reaction at 260 °C.

Based on our in situ DRIFTS, in situ and operando XANES and EXAFS experiments, and relevant research findings[25], we put forth a reaction mechanism elucidating the process of $CO_2$ hydrogenation to methanol over Cu/$MoS_2$@$SiO_2$ (Supplementary Fig. 33). Initially, $CO_2$ exhibits preferential chemisorption on the in-plane Sv, leading to the dissociation and generation of CO* and O* species. Subsequently, CO* undergoes stepwise hydrogenation, resulting in the formation of CHO*, $CH_2O$*, and $CH_3O$* intermediates, ultimately leading to the formation of methanol[25]. By decorating the catalyst surface with the atomically dispersed Cu−S, the generation of CO* and the stepwise hydrogenation to $CH_3O$* and $CH_3OH$ can be promoted due to the increase of Sv and the positive hydrogen activation effect kindled by Cu involvement.

## DFT calculations for the role of strain

Finally, DFT calculations were conducted to reveal how introduced curvature in $MoS_2$ basal plane can indeed impact the Sv formation and $CO_2$ hydrogenation performance. As an initial simplified model, strained $MoS_2$ monolayer film consisting of an S−Mo−S tri-layer can serve as a platform to study how the strain influences the Sv formation in basal planes. The lattice strain we introduced in the monolayer film is from −5% to 15%. Firstly, our calculations confirmed that unstrained film has the lowest energy compared to the strained ones (Supplementary Fig. 34). Sv formation energies were calculated by removing one S atom from the strained films (Fig. 7a; blue dot curve). The compression of the film facilitates the removal of one S atom from the $MoS_2$ basal plane as indicated by the decreasing Sv formation energy. The tensile strain up to 8% increased the Sv formation energy (up to 2.70 eV). Further increasing the tensile strain ($\varepsilon \geq$ 9%) led to rapid decrease of Sv formation

energies and the reconstruction of the film (Fig. 7a; green square curve), where S atom below vacancy moved to the center of triangle formed by 3 Mo atoms.

As have shown in our experiments, $MoS_2$ formed a fullerene-like sphere which introduced strain into $MoS_2$. In simulations herein, we investigated the effect of curvature on strain and reactivity of $MoS_2$ using $MoS_2$ nanotube models, to show how the curvature affects Sv formation and hydrogen dissociation. The elimination of strain along nanotube axis was ensured by optimization of vector $\vec{c}$ length (Supplementary Fig. 35). The nanotube interior and exterior surfaces bear compressive and tensile strain (Fig. 7a; orange and red curves), respectively, which can be measured through S−S bond lengths on these surfaces (Supplementary Figs. 36, 37, Supplementary Table 7). Similar to strained $MoS_2$ films, the high compressive strain in the interior surface of narrow $MoS_2$ nanotubes with high curvature facilitated the formation of Sv. In turn, the low tensile strain in $MoS_2$ nanotubes with large diameter made the vacancy formation energy more endothermic (Fig. 7a).

We further examined catalytic properties of strained $MoS_2$. The compressively strained $MoS_2$, which facilitated Sv formation, turned out to have higher $H_2$ dissociation Gibbs free energy compared to unstrained films (Fig. 7b). The hindering effect of strain on $H_2$ activation intensified as the compressive strain increased. In contrast, stretching the film facilitated $H_2$ dissociation with a $G_{2H}$ of −0.16 eV at 5% tensile strain. Akin to strained $MoS_2$ films, tensile strain in the $MoS_2$ nanotube exterior surface facilitates $H_2$ dissociation. For example, the (30,0) nanotube exterior bears 6.4% tensile strain and features $G_{2H}$ of −0.14 eV. On the other hand, low compressive strain in large $MoS_2$ nanotube interior surface made $H_2$ dissociation more endothermic. However, when the diameter became smaller than c.a. 1.9 nm, the endothermicity for $H_2$ dissociation decreased with higher compressive strain caused by higher curvature. Note that both the vacancy formation energy and the dissociative $H_2$ adsorption energy calculated for $MoS_2$ nanotubes approach the values for strained films when the strain is low, suggesting that strained $MoS_2$ films accurately mimic the chemical properties of curved $MoS_2$ in the low-strain region.

Inspired by the similarity between reactivities of strained films and curved $MoS_2$, we further investigated the $CO_2$ reduction reaction on $MoS_2$ films bearing −2% and 2% of strain-like $MoS_2$ spheres in our experiments. Previous studies established that two adjacent Sv are the active site for $CO_2$ reduction to methanol and calculated reaction energy profile for this process, which serves here as the reference to demonstrate the effect of strain on the catalytic activity[25]. To facilitate the comparison between the current work and the previous study, the configurations of intermediates on strained films remained the same as on the non-strained film. Compared to zero strain, 2% of compressive strain destabilized the adsorption of all reaction intermediates, whereas tensile strain in $MoS_2$ stabilized all the intermediates and made the dissociation of $H_2$ (−0.04 eV) and adsorption of $CO_2$ (−0.26 eV) exothermic (Fig. 7c). The stabilizing and destabilizing of reaction intermediates due to lattice expansion and contraction are commonly observed and often attributed to the catalyst $d$ band shift[61]. However, our results show that the effect of the strain on stability of reaction intermediates varied significantly among the adsorbed species. O had the smallest binding energy changes (−0.03 eV/+0.10 eV) after applying 2% of tensile strain or compressive strain, and (H + H + $CO_2$)* had the largest changes (−0.56 eV/+0.65 eV). We adopted the energetic span model to analyze how strain affected the overall reaction performance by dividing the reaction mechanism into two parts, i.e., $CO_2$ reduction to $CH_3OH$ and the hydrogenation of the O generated by $CO_2$ dissociation[25]. The rate-determining states for methanol formation are ($CH_2O$ + O)* and (TS1)*, the difference in free energies of which defines the first energetic span (Fig. 7d). In ($CH_2O$ + O)*, both C and O atoms of $CH_2O$* were bound to Mo atoms.

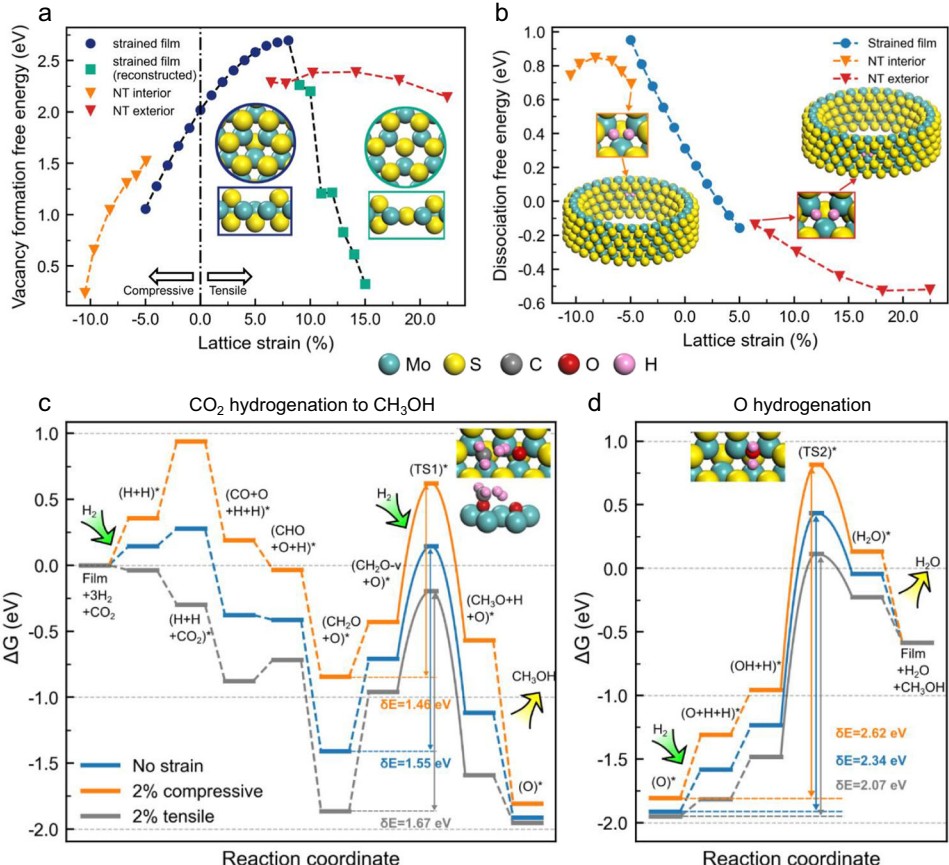

**Fig. 7 | DFT calculations of the reaction mechanisms of $CO_2$ hydrogenation on the strained $MoS_2$. a** Sv formation energies of strained $MoS_2$ films and $MoS_2$ nanotube (NT) with interior and exterior surfaces. The insets show top and side views of two distinct relaxed structures of $MoS_2$ films with one Sv at low and high strain levels. **b** Energies of dissociative adsorption of $H_2$ on Sv of strained films and $MoS_2$ nanotubes. The strain values of nanotube interior and exterior are determined by their average S-S bond lengths (Supplementary Table 7). The insets show the structures of $H_2$ dissociative adsorption on 3.37 nm wide (30,0) nanotubes. **c, d** Gibbs free energy reaction profiles for $CO_2$ hydrogenation to $CH_3OH$ (**c**) and hydrogenation of O (**d**) formed on $MoS_2$ films with no strain (blue line), compressive (orange line), and tensile (gray line) strains. Insets are structures of the two most important transition states: $CH_2O^*$ hydrogenation to $CH_3O^*$ with $H_2$ molecule as the attacking species (TS1) and O hydrogenation to $H_2O$ (TS2). Energetic spans (δE) of these profiles are also given in the figure.

After C-Mo bond broke in this intermediate to form the $(CH_2O\text{-}v + O)^*$ structure, an $H_2$ molecule attacked the C atom through $TS_1$. By applying 2% of compressive strain, the energetic span (δE) for methanol formation decreased from 1.55 eV to 1.46 eV; on the other hand, the 2% tensile strain increased the energetic span to 1.67 eV, which may be explained by lower reactivity of strongly bound intermediates. However, the energetic span for O hydrogenation increased from 2.34 eV to 2.62 eV by 2% compressive strain (Fig. 7d). As indicated by the reduced energetic span, 2% tensile strain facilitates the hydrogenation of O, preventing the catalyst from deactivation through oxidation. Such effect of strain on the rate of O hydrogenation can be attributed to longer H−OH distances in the TS2 on films with more compressive strain (Supplementary Tables 8, 9). Note that other transition states were calculated to be less important for the reaction kinetics in a previous study[25]. Therefore, the results suggest a higher methanol formation rate on the interior of $MoS_2$ spheres enduring compressive strain and a higher O hydrogenation rate on the exterior of $MoS_2$ spheres with tensile strain, which overall gives a better catalytic performance for $CO_2$ hydrogenation.

## Discussion

In summary, we report the selective exposure and activation of the inert basal plane of 2H-$MoS_2$ for $CO_2$ hydrogenation to methanol by physically constrained sulfidation of ultrafine $MoO_2$ nanocores to hollow few-layer $MoS_2$ sphere within mesoporous silica. The spherical curvature of $MoS_2$ enables the generation of strain and Sv in originally inert basal plane. The fullerene-like $MoS_2$ can selectively expose more in-plane Sv and simultaneously reduce the exposure of edge Sv, and thus are extremely conducive to methanol synthesis from $CO_2$ hydrogenation. By further anchoring atomic Cu to facilitate $H_2$ activation, the obtained Cu/$MoS_2$@$SiO_2$ can achieve an excellent specific methanol yield of 6.11 mol $mol_{Mo}^{-1}$ $h^{-1}$ with high methanol selectivity of 72.5% at 260 °C, 5 MPa, and 24,000 mL $g_{cat.}^{-1}$ $h^{-1}$ in $CO_2$ hydrogenation, which significantly surpassed its counterparts without this well-designed few-layer fullerene-like structure. This Cu/$MoS_2$@$SiO_2$ catalyst exhibited excellent stability during the reaction and its silica-encapsulated hollow $MoS_2$ core structure remained unchanged. Furthermore, the reaction mechanism and the promotional roles of atomic Cu are investigated by in situ DRIFTS and in situ XAS. DFT calculations reveal that the compressive strain facilitates Sv formation and $CO_2$ hydrogenation while the tensile strain accelerates the regeneration of active sites, validating the critical role of strain. Considering that introduction of strain into $MoS_2$ and the selective exposure of in-plane Sv are both important for improving and regulating the catalytic performance of various thermal/electro catalytic reactions, we believe the developed rigorous synthesis strategy of fullerene-like $MoS_2$ reactor could guide the design of more efficient catalysts beyond $CO_2$ hydrogenation in the future.

## Methods

### Synthesis of $MoO_2$ nanocores

$MoO_2$ nanocores were prepared using a straightforward one-pot hydrothermal synthesis method, following our previously reported procedure with minor adjustments[39]. Briefly, 150 mg of AMT was dissolved in 22.0 mL of deionized water, and 10 mL of ethanol was subsequently added. Later, 0.5 g of PVP was introduced into the solution, which was then stirred at room temperature for 30 minutes. The resulting mixture was transferred into a Teflon-lined stainless-steel autoclave with a capacity of 50 mL, and subjected to hydrothermal treatment in an electric oven at 180 °C for 16 hours. After that, the autoclave was allowed to cool to ambient temperature, and the dark precipitate (referred to as $MoO_2$ nanocores) was collected by centrifugation. The $MoO_2$ nanocores were then washed with ethanol-acetone cosolvent multiple times and dried at 60 °C overnight in a vacuum drying oven. In addition, by maintaining other parameters constant and varying the initial amount of AMT (100 mg, 150 mg, 200 mg, 250 mg, and 300 mg), $MoO_2$ nanoparticles with a larger size (e.g., average particle size of 31 nm, 47 nm, 66 nm, 112 nm, and 147 nm) could also be synthesized.

### Synthesis of $MoO_2@SiO_2$

$MoO_2@SiO_2$ were synthesized using a modified version of our previous method involving the hydrolysis and condensation of tetraethyl orthosilicate[37,38]. To begin, 120 mg of $MoO_2$ nanocores were added into a mixed solvent (132 mL water and 80 mL methanol), followed by 20 minutes of sonication. Subsequently, 2.2 mL of a 25% cetyltrimethylammonium chloride (CTAC) solution and 800 mg of 2-methylimidazole were added to the dispersion, and the resulting mixture was stirred for 30 minutes. After that, 1.6 mL of tetraethyl orthosilicate (TEOS) was added to the solution during continuing stirring, and the resultant mixture was stirred for an additional 3 hours at room temperature. The resulting gray solid was isolated through centrifugation, followed by washing with a cosolvent of acetone and ethanol, and finally dried at 50 °C overnight in a vacuum drying oven, resulting in the formation of $MoO_2@SiO_2$. By reducing the reaction time for silica deposition to 0.5, 1, and 2 hours, $MoO_2@SiO_2$ spheres with varying thicknesses of the mesoporous silica shell could be obtained.

### Synthesis of $MoS_2@SiO_2$

The $MoO_2@SiO_2$ sample obtained previously was transformed into $MoS_2@SiO_2$ as follows: 100 mg of $MoO_2@SiO_2$ was dispersed in 32 mL of water and subjected to 30 minutes of sonication, after which 300 mg of thioacetamide (TAA) was added to the dispersion. The solution was then transferred into a 50 mL Teflon-lined stainless-steel autoclave container and subjected to hydrothermal treatment at 200 °C for 24 hours. Subsequently, the autoclave was cooled to room temperature in a fume hood, and the resulting precipitate was harvested by centrifugation, followed by washing with ethanol and drying at 60 °C overnight in a vacuum drying oven. Finally, the obtained sample was calcined in an Ar atmosphere at 700 °C for 2 hours with a ramping rate of 5 °C min⁻¹ to yield $MoS_2@SiO_2$.

### Synthesis of $Cu/MoS_2@SiO_2$

$Cu/MoS_2@SiO_2$ were synthesized using a modified version of our previously reported method[41]. Firstly, 0.475 mL of copper acetate precursor ethanol solution (0.05 M) was added to 100 mg of $MoS_2@SiO_2$ in a glass bottle with 2 min of sonication. Then the glass bottle was placed in 80 °C oven for 1 h for drying. Then the obtained samples were calcined in Ar at 500 °C for 3 h with a ramping rate of 5 °C min⁻¹ to yield $Cu/MoS_2@SiO_2$. Unless specified, the loading amount of Cu in the $Cu/MoS_2@SiO_2$ is 1.5 wt.%. In addition, $Cu/MoS_2@SiO_2$ with different Cu loadings could also be synthesized by changing the initial copper acetate precursor amount while other parameters remained unchanged.

### Characterization methods

The structures of catalysts were characterized using FESEM, TEM, HRTEM equipped with energy-dispersive X-ray (EDX) elemental analysis, XRD, inductively coupled plasma optical emission spectrometry (ICP-OES), $N_2$ physisorption, XPS, EPR, Raman spectrometer and $H_2$-TPD.

### Hydrogenation reaction of $CO_2$ to methanol

The catalytic activity evaluations were conducted using a PID Eng&Tech four-channel high-pressure fixed-bed flow reactor. For each experiment, 150 mg of prepared sample was introduced into the lower section of the quartz reaction tube with an inner diameter of 4 mm. For a fair comparison, 52 mg $MoS_2$-Com, $MoS_2$-NPs, and $MoS_2$-HT were mixed with 98 mg mesoporous silica, respectively, to achieve an equivalent $MoS_2$ mass for all catalysts used in $CO_2$ hydrogenation experiments. Subsequently, the quartz tube was enclosed within the stainless-steel reaction tube and tightly sealed. Prior to the catalytic tests, the pretreatment process was conducted under a continuous flow of $H_2$ (20 mL min⁻¹) at 300 °C and atmospheric pressure for a duration of 3 hours. Upon reaching the target reaction temperature, the reaction gas ($H_2$:$CO_2$:Ar= 76/19/5, Ar as an internal standard, flow rate of 20 mL min⁻¹) was introduced into the reactor. The precise control of $CO_2$, $H_2$, and Ar compositions was achieved using mass flow controllers. The reactor pressure was elevated to 5 MPa through an automated digital pressure regulator. To prevent gaseous product condensation, the outlet gas line was maintained at 180 °C. The effluent was continuously sampled and analyzed online using an automated gas chromatography (GC, Agilent 8890 A) equipped with a thermal conductivity detector (TCD), flame ionization detector (FID), and Agilent HP-Poraplot Q and Restek ShinCarbon chromatographic columns.

### DFT calculations

Spin-polarized density functional theory (DFT) calculations were carried out using Vienna Ab initio Simulation Package (VASP)[62–64]. The exchanged-correlation contribution to the total energy was treated with the Perdew-Burke-Ernzerhof (PBE) implementation of generalized gradient approximation (GGA)[65]. Projector Augmented-Wave was adopted to describe the core-valance interaction[66,67], and the cut-off energy for the plane-wave-basis set was 400 eV. The DFT-D3 method of Grimme with zero-damping function was applied to take into dispersion interactions[68]. A Γ point and 1×1×2 Monkhorst-Pack sampling of the first Brillouin zone were applied to calculations of $MoS_2$ films and of nanotubes, respectively[69]. Convergence criteria for self-consistent field calculations and geometry optimizations were $1 \times 10^{-5}$ eV and 0.03 eV/Å, respectively.

A 5 × 5 $MoS_2$ film was used to study the effect of strain on Sv formation and $H_2$ dissociation. The $CO_2$ reduction reaction profile was studied on a 6 × 6 film with two Sv as suggested in earlier studies[25]. Sv formation energy ($E_{Suf}$) is defined as the energy required to reduce the pristine nanostructured $MoS_2$ with $H_2$ to form $H_2S$ and $MoS_2$ with one Sv:

$$E_{suf} = E(H_2S) + E(Mo_xS_{2x-1}) - E(H_2) - E(Mo_xS_{2x}) \qquad (1)$$

where $E(H_2S)$ and $E(H_2)$ are the energy of gas-phase $H_2S$ and $H_2$, $E(Mo_xS_{2x-1})$ is the energies of $MoS_2$ with one Sv, and $E(Mo_xS_{2x})$ is the energy of pristine nanostructured $MoS_2$. The energy of dissociative adsorption of $H_2$ ($E_{2H}$) is defined in Eq. (2):

$$E_{2H} = E(2H + Mo_xS_{2x-1}) - E(H_2) - E(Mo_xS_{2x-1}) \qquad (2)$$

where $E(2H + Mo_xS_{2x-1})$ is the energy of $MoS_2$ with one vacancy and two H atoms adsorbed. The relative energy is defined as the total energy change with reference to the unreacted species, i.e., film $+3H_2(g)+CO_2(g)$. $H_2(g)$ and $CO_2(g)$ are single molecules in the gas-phase. Biaxial lattice strain ($\varepsilon$) of the $MoS_2$ film is defined as Eq. (3):

$$\varepsilon = \frac{(a_s - a_0)}{a_0} \quad (3)$$

where $a_0$ is the lattice constant of the unstrained $MoS_2$ film obtained from the optimization of bulk $MoS_2$, and $a_s$ is the lattice constant of a strained film. Note that the total energy of the film achieves lowest value at 0 strain. Positive lattice strain values indicate tensile strain while negative values mean compressive strain. Sv formation energy and relative energy were corrected to get the free energy. Gibbs energy for the adsorption system was calculated using the following Eq. (4):

$$G = E_{DFT} + ZPVE + H_{vib} - TS_{vib} \quad (4)$$

where $E_{DFT}$ is the DFT total energy. Zero-point vibrational energy ($ZPVE$), contributions to enthalpy ($H_{vib}$), and entropy ($S_{vib}$) due to vibrations were calculated based on the vibrational frequencies. Vibrations of the adsorbates were evaluated using the finite differences approach with ±0.015 Å displacements, while the film and nanotube atoms were kept fixed. Gibbs energy corrections for gas-phase molecules were treated in the ideal gas approximation. Reaction energies were corrected based on the reaction conditions (533 K, 37.5 bar for $H_2$, 12.5 bar for $CO_2$, 0.75 bar for $CH_3OH$ and $H_2O$), and the Sv formation energies were corrected based on the reduction conditions (573 K, 1 bar for $H_2$, 0.01 bar for $H_2S$)[70]. An additional 0.44 eV free energy correction to gas-phase $CO_2$ was applied to amend thermochemical reaction energies of $CO_2$ reduction[25,71]. Furthermore, information on other DFT calculations for $Cu/MoS_2$ system can be found directly in Supplementary Information.

## Data availability

Relevant data supporting the key findings of this study are available within the article and the Supplementary Information file. All raw data generated during the current study are available from the corresponding authors upon request.

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

## Acknowledgements

The authors gratefully acknowledge the financial support provided by the National University of Singapore, and the National Research Foundation (NRF), Prime Minister's Office, Singapore, under its Campus for Research Excellence and Technological Enterprise (CREATE C4T Program) as well as from Agency for Science, Technology and Research (A*Star) through the grant LCERFI01-0033|U2102d2006. Computational work was performed using resources of the National Supercomputing Centre, Singapore.

## Author contributions

S.Z. designed and conducted most of the experimental work, data analysis, and interpretation, and composed the initial draft of the manuscript. W.M. performed the DFT simulations and drafted the DFT part under S.M.K. and U.A.'s supervision, who revised the DFT part. S.X.

carried out XAS experiments and assisted with XAS data analysis. M.K. assisted with XAS experiments, and data analysis and reviewed the manuscript. H.C.Z. conceived the project idea, supervised the research implementation, data analysis, curation, and interpretation, and reviewed and revised the manuscript.

## Competing interests

The authors declare no competing interests.
