## [Peer Review File · Nature Communications]

Strained Few-Layer MoS₂ with Atomic Copper and Selectively Exposed In-plane Sulfur Vacancies for CO₂ Hydrogenation to MethanolREVIEWER COMMENTS

Reviewer #1 (Remarks to the Author):

In this work, Dr. Zeng, Kozlov and their co-workers reported fabrication of a delicate strained Cu/MoS₂ catalyst through silica constraint synthesis strategy. The in-plane sulfur vacancy are selectively enriched via growing bended MoS₂ layers inside silica pores. Comprehensive evidence of microscopy imaging and elemental analysis, in situ infrared spectroscopy and XAS provide systematic details about the strained-MoS₂ formation as well as Cu atoms anchor location. In general consideration I would recommend the publication of this innovative discovery after following questions have been properly answered.

1. During the sulfidation of loosely packed MoO₂ cores inside SiO₂, the resulted MoS₂ are not fully converted into strained layered structure (fullerene-like) but remarkable part remains in randomly stacked small MoS₂ pieces inside the fullerene-MoS₂ encapsulation, which inherits MoO₂ morphology (see Fig. 3a2,a5 and b2,b5). These small MoS₂ pieces are not as seriously bended as outside fullerene-MoS₂ layers, then more edges sites are expected to expose. How to evaluate contribution of these unique MoS₂ pieces to the reaction?
2. In Figure 3 a5 and b5, elemental mappings are in different morphologies with their corresponding HAADF images, are they actually in different positions or MoS₂@SiO₂ spheres re-arranged during EDS mapping?
3. From Figures. 12, the as-formed fullerene MoS₂ layers have generally larger diameter than their original MoO₂ precursor NPs. This suggests that the constrained sulfurization was companied with mass diffusion process of Mo species. It thus leads us to speculate that the porosity of silica shell is one important factor to facilitate the growth of strained-MoS₂ layers. Therefore, besides the two factors, silica shell thickness and MoO₂ diameter as author summarized, I would suggest the silica porosity as the third parameter to be considered during fabricating MoS₂@SiO₂. For example, for MoO₂@SiO₂ with MoO₂ larger than 66nm, would it be possible to obtain well strained-MoS₂ layers without leaving unsulfurized MoO₂ core through optimizing the silica porosity and encapsulation thickness?
4. In XPS analysis, it is unclear that Mo+6 is only abundant in the primary samples? (MoS₂@SiO₂ and Cu/MoS₂@SiO₂), but not observed on those rare MoS₂ counterparts (HT-, NPs-, Com-)? Then why MoS₂ encapsulated within protection of SiO₂ shell is more prone to be oxidized to high valence than those bare MoS₂? A more rational explanation is necessary.
5. From Table S2, there is apparent existence of Cu-Cu coordination besides Cu-S ones, then why author get conclusion that Cu in Cu/MoS₂@SiO₂ sample is in “atomic dispersion of Cu species”?
6. Known that Cu-based catalyst itself shows good performance in Methanol synthesis, to uncouple the contribution of the synergy between Cu decoration and in-plane S vacancies of strained-MoS₂, it is necessary to provide the performance of Cu@SiO₂ upon CO₂ reduction to be compared with activity of Cu/MoS₂@SiO₂ sample at the same fabrication protocol with Cu/MoS₂@SiO₂.

7. format details should be carefully confirmed. e.g. journal aberration are not unified ref 23, ref

Reviewer #2 (Remarks to the Author):

The authors report the fabrication of strained MoS₂ hollow spheres with atomic Cu incorporation as an active catalyst for methanol production from CO₂ hydrogenation. As main results, the authors claim that (1) the spherical curvature of FL-MoS₂ encapsulated by SiO₂ shell enabled the formation of strain and S-vacancy in the basal plane, (2) the atomic Cu species promotes the methanol production performance, and (3) the strains, depending on being tensile or compressive, can facilitate S-vacancy formation or enhance H₂ dissociation. The reaction is certainly an important topic and the efforts conducted by the authors such as detailed characterization, catalytic test, in-situ DRIFT measurements and DFT calculations seem to be sound. However, the novelties of the present work are limited as S-vacancy-rich MoS₂ for CO₂ hydrogenation and fabrication of catalysts with strains have been intensively studied. There are other deficiencies that lead the reviewer to think that this work does not meet the standards of Nat. Commun. Below are some specific thoughts.

1. It is understood that the SiO₂ shell is employed here to confine the transformation of the MoO₃ core to form FL-MoS₂ hollow spheres. However, the reviewer wonders if it is necessary to keep the SiO₂ shell for catalysis as it may block the active surface and hinder the transport of reactants/products. If the SiO₂ shell was to be removed, does it affect the activity?

2. One of the most concerning aspects is the chemical state of Cu and its role in catalysis. The author performed XAS measurements and EXAFS fitting and claimed that (1) Cu is bonded to S and (2) Mo atom is not replaced by Cu in reduced Cu/MoS₂@SiO₂ catalyst. The former is clear from Figure 4e while the latter is not well supported by the experimental data. As the FL-MoS₂ exhibits S vacancies, it can be expected that the CN of Mo is significantly lower than that in the bulk structure. Therefore, the comparison of CN of Cu with that of Mo in bulk MoS₂ is not valid. The author should perform EXAFS fitting for Mo and compare Cu with it to clarify the chemical environment of Cu. The model shown in Figure 4f is not convincing as Cu single atoms located in the MoS₂ lattice could exhibit a similar EXAFS spectrum.

3. In addition, in the DFT calculations, the author did not take the atomic Cu species into account when constructing the models. It should be clarified if the role of Cu is only to induce a higher number of S vacancies in MoS₂ or if it can optimize the reaction energetics in the CO₂ hydrogenation reactions.

4. The in-situ DRIFT measurements provide limited insights on the benefitting role of Cu as the IR bands intensity of intermediates and CH₃OH are expected to be greater if the catalyst already exhibits higher CH₃OH production rates from CO₂ hydrogenation.

5. The author lists in the reference section are highly inconsistent.

REVIEWER COMMENTS

Reviewer #1 (Remarks to the Author):

In this work, Dr. Zeng, Kozlov and their co-workers reported fabrication of a delicate strained Cu/MoS₂ catalyst through silica constraint synthesis strategy. The in-plane sulfur vacancy are selectively enriched via growing bended MoS₂ layers inside silica pores. Comprehensive evidence of microscopy imaging and elemental analysis, in situ infrared spectroscopy and XAS provide systematic details about the strained-MoS₂ formation as well as Cu atoms anchor location. In general consideration I would recommend the publication of this innovative discovery after following questions have been properly answered.

Reply: Thank you for the positive feedback regarding the impact and innovation of our research. We have taken all these very valuable suggestions and have supplemented relevant experiments in this revised manuscript.

1. During the sulfidation of loosely packed MoO₂ cores inside SiO₂, the resulted MoS₂ are not fully converted into strained layered structure (fullerene-like) but remarkable part remains in randomly stacked small MoS₂ pieces inside the fullerene-MoS₂ encapsulation, which inherits MoO₂ morphology (see Fig. 3a2,a5 and b2,b5). These small MoS₂ pieces are not as seriously bended as outside fullerene-MoS₂ layers, then more edges sites are expected to expose. How to evaluate contribution of these unique MoS₂ pieces to the reaction?

Reply: Thank you for the good question. Our effort in this work was to reduce the exposure of edge Sv as much as possible but not 100%. It has also been shown that this unique fullerene-like structure does significantly reduce the production of methane. Specifically, for samples without this fullerene-like structure (MoS₂-HT and MoS₂-NPs), the methane selectivity was 50% and 45%, respectively. However, our MoS₂@SiO₂ with this structure has a noticeably lower methane selectivity of 23% under the same conditions, we can therefore conclude that the MoS₂@SiO₂ sample with fullerene-like structure can selectively expose more in-plane Sv, simultaneously reduce the exposure of edge Sv. This proves that our designed structure is effective. However, after the sulfidation of MoO₂@SiO₂ to MoS₂@SiO₂, not all MoO₂ cores could be fully converted into a perfect fullerene-like structure despite our extensive optimization experiments. These small MoS₂ pieces randomly exposed both in-plane and edge Sv, so there is still a small amount of methane in the product.

In fact, only the few-layered 2H-phase MoS₂ can catalyze CO₂ hydrogenation to methanol. Although some methods have been reported for the preparation of fullerene-like MoS₂. These synthesized MoS₂ are essentially multilayer structures and are not suitable for methanol synthesis reactions. The highlight of our synthesis process is that due to the small size of the synthesized MoO₂ cores, they can be converted into MoS₂ with both few-layered and fullerene-like structure during the subsequent constrained sulfidation process. This unique structure can selectively expose more in-plane Sv, which is beneficial for methanol synthesis. To the best of our knowledge, the preparation of MoS₂ with both few-

layered and fullerene-like structure has not been reported. We will continue to work on how to fully convert $\text{MoO}_2@\text{SiO}_2$ to $\text{MoS}_2@\text{SiO}_2$ without randomly stacked MoS_2 pieces inside MoS_2 hollow sphere.

2. In Figure 3 a5 and b5, elemental mappings are in different morphologies with their corresponding HAADF images, are they actually in different positions or $\text{MoS}_2@\text{SiO}_2$ spheres re-arranged during EDS mapping?

Reply: Thank you for your very careful reading of our manuscript. Yes, HAADF and EDS mapping were indeed different positions. After we took the HAADF images, opened the EDS software and adjusted the parameters, we found that our spherical nanoparticles tend to drift around under the electron beam. Therefore, we had to find another more stable location to shoot the EDS mapping for the best display. In this regard, we have revised the caption of **Fig. 3.** and corresponding description in the manuscript.

3. From Figures. 12, the as-formed fullerene MoS_2 layers have generally larger diameter than their original MoO_2 precursor NPs. This suggests that the constrained sulfuration was accompanied with mass diffusion process of Mo species. It thus leads us to speculate that the porosity of silica shell is one important factor to facilitate the growth of strained- MoS_2 layers. Therefore, besides the two factors, silica shell thickness and MoO_2 diameter as author summarized, I would suggest the silica porosity as the third parameter to be considered during fabricating $\text{MoS}_2@\text{SiO}_2$. For example, for $\text{MoO}_2@\text{SiO}_2$ with MoO_2 larger than 66nm, would it be possible to obtain well strained- MoS_2 layers without leaving unsulfurized MoO_2 core through optimizing the silica porosity and encapsulation thickness?

Reply: Thank you for your suggestion. We strongly agree with your comments about MoO_2 larger than 66 nm can be fully sulfurized by adjusting the porosity of silica. In fact, we also found that the larger diameter MoO_2 could also be fully sulfurized by simply extending the sulfidation time to 36 h. However, it is noteworthy that the focus of our paper is not to sulfurize large diameter MoO_2 . As we have mentioned before, only the few-layered MoS_2 can catalyze CO_2 hydrogenation to methanol. Due to the extremely small size of the synthesized MoO_2 cores and constrained sulfidation process, the MoS_2 formed does not aggregate and MoO_2 cores can be converted into few-layered MoS_2 . Although large diameter MoO_2 (66 nm) can also be sulfurized (**Supplementary Fig. 13**), the MoS_2 formed is multilayered (6–8 layers), which does not facilitate the exposure of sufficient Sv. We have also tested the performance of this catalyst (**Supplementary Table 4**). The results show that it has a significantly lower CO_2 conversion. Therefore, in a subsequent study we used MoO_2 with small diameter to prepare $\text{MoS}_2@\text{SiO}_2$; the main findings of this work are based on our study using the small-sized MoS_2 cores.

Supplementary Fig. 13. TEM images of MoS₂@SiO₂-66nm (MoO₂ with an average diameter of 66 nm as Mo source).

Supplementary Table 4. Catalytic performance of different samples in CO₂ hydrogenation to methanol.

Sample names	Reaction conditions	Conversion (%)	MeOH Selectivity (%)	CO Selectivity (%)	CH ₄ Selectivity (%)
MoS ₂ @SiO ₂	260 °C, 5 MPa,	11.1	52.2	22.5	24.6
MoS ₂ @SiO ₂ -66nm	GHSV=8000 mL·g _{cat.} ⁻¹ ·h ⁻¹	7.5	50.2	23.6	26.2

MoS₂@SiO₂-66nm: MoO₂ with an average diameter of 66 nm as Mo source.

4. In XPS analysis, it is unclear that Mo+6 is only abundant in the primary samples? (MoS₂@SiO₂ and Cu/MoS₂@SiO₂), but not observed on those rare MoS₂ counterparts (HT-, NPs-, Com-)? Then why MoS₂ encapsulated within protection of SiO₂ shell is more prone to be oxidized to high valence than those bare MoS₂? A more rational explanation is necessary.

Reply: Thank you for the good questions. Due to the extremely small size of the synthesized MoO₂ cores and constrained sulfidation process, the MoS₂ formed does not aggregate and MoO₂ cores can be converted largely into few-layered MoS₂. In contrast, for MoS₂ counterparts (MoS₂-HT, MoS₂-NPs, MoS₂-Com), they have a multilayer or thick-layered nanosheet structure. It is clear that the highly exposed few-layered MoS₂ of MoS₂@SiO₂ and Cu/MoS₂@SiO₂ are more active for hydrogenation but at the same time more susceptible to be oxidized to higher valence. For MoS₂ samples with a thick nanosheet structure (MoS₂-HT, MoS₂-NPs, MoS₂-Com), most of the Mo are located in the middle of the nanosheet layer, which makes them less susceptible to oxidation. Therefore,

Mo⁺⁶ content in MoS₂@SiO₂ and Cu/MoS₂@SiO₂ with few-layer MoS₂ structure is higher when compared to multi-layered samples (MoS₂-HT, MoS₂-NPs, MoS₂-Com).

5. From Table S2, there is apparent existence of Cu-Cu coordination besides Cu-S ones, then why author get conclusion that Cu in Cu/MoS₂@SiO₂ sample is in “atomic dispersion of Cu species”?

Reply: We conclude that 93% of Cu in Cu/MoS₂@SiO₂ sample is mainly atomically dispersed on MoS₂ through the Cu-S bond based on linear combination fitting (**Supplementary Table 4**). However, small amounts of Cu nanoparticles/clusters are still present in the sample. We have added the corresponding description in the manuscript.

6. Known that Cu-based catalyst itself shows good performance in Methanol synthesis, to uncouple the contribution of the synergy between Cu decoration and in-plane S vacancies of strained-MoS₂, it is necessary to provide the performance of Cu@SiO₂ upon CO₂ reduction to be compared with activity of Cu/MoS₂@SiO₂ sample at the same fabrication protocol with Cu/MoS₂@SiO₂.

Reply: Thanks for your suggestion! We have added the catalytic performance of Cu@SiO₂ (1.5wt.% Cu) based on your suggestion. The results show a low activity of the Cu@SiO₂ with only 2.5% CO₂ conversion and 50.4% methanol selectivity, which is consistent with our recent published work.¹ Indeed, it is well known that Cu-based catalysts have been widely used for methanol synthesis in recent years. However, the Cu content, the choice of supports and the interfacial environment play a crucial role in the reaction.¹ In general, Cu/ZnO or Cu/ZrO₂ catalysts have the best activity. In this paper, the role of Cu is mainly used to promote the formation of S vacancies and activate hydrogen based on our characterization (EPR spectra, H₂-TPD) and our new DFT calculations (**Figure 6**).

Supplementary Table 4. Catalytic performance of Cu@SiO₂ and Cu/MoS₂@SiO₂ in CO₂ hydrogenation to methanol.

Sample names	Reaction conditions	Conversion (%)	MeOH Selectivity (%)	CO Selectivity (%)	CH ₄ Selectivity (%)
Cu@SiO ₂	Cu:1.5wt%, 260 °C, 5 MPa,	2.5	50.4	49.6	-
Cu/MoS ₂ @SiO ₂	GHSV=8000 mL·g _{cat.} ⁻¹ ·h ⁻¹	12.8	59.2	23.9	16.4

1. Chen C, Kosari M, Xi S, Zeng H. C., et al. Optimizing the Interfacial Environment of Triphasic ZnO-Cu-ZrO₂ Confined inside Mesoporous Silica Spheres for Enhancing CO₂ Hydrogenation to Methanol. ACS ES&T Engineering, 2023, 3(5): 638-650.

7. format details should be carefully confirmed. e.g. journal aberration are not unified ref 23, ref.

Reply: Thank you for pointing out these errors. We have carefully revised and confirmed the format of the references accordingly.

Reviewer #2 (Remarks to the Author):

The authors report the fabrication of strained MoS₂ hollow spheres with atomic Cu incorporation as an active catalyst for methanol production from CO₂ hydrogenation. As main results, the authors claim that (1) the spherical curvature of FL-MoS₂ encapsulated by SiO₂ shell enabled the formation of strain and S-vacancy in the basal plane, (2) the atomic Cu species promotes the methanol production performance, and (3) the strains, depending on being tensile or compressive, can facilitate S-vacancy formation or enhance H₂ dissociation. The reaction is certainly an important topic and the efforts conducted by the authors such as detailed characterization, catalytic test, in-situ DRIFT measurements and DFT calculations seem to be sound. However, the novelties of the present work are limited as S-vacancy-rich MoS₂ for CO₂ hydrogenation and fabrication of catalysts with strains have been intensively studied. There are other deficiencies that lead the reviewer to think that this work does not meet the standards of Nat. Commun. Below are some specific thoughts.

Reply: We sincerely thank the Reviewer for appreciating the experimental and theoretical aspects of our research work. We also thank the Reviewer for the constructive suggestions on our research. In response to the deficiencies identified by the Reviewer, we have taken all these very valuable suggestions. In particular, we have carried out relevant experiments, performed new DFT calculations and made extensive changes to the article. To better highlight these key aspects, which may not have been recognized in our original contribution, we have amended the title, abstract, and the corresponding section of the main text.

To further highlight and demonstrate the innovativeness of our work, we summarise the main achievements of the present study as follows:

(1) When compared to the widely investigated Cu-based catalysts, In₂O₃ and ZnO/ZrO₂-based oxide catalysts, we believe that very few studies have been reported on MoS₂ in CO₂ hydrogenation since the first paper in 2021.¹ The main reason is the more complex structure of sulfides compared to metal oxide catalysts and the structure of sulfide has a crucial effect on catalytic activity. MoS₂ has three phases (1T, 2H, and 3R), various layered structures (multilayer, few-layer and single-layer), and two types of active sites (in-plane and edge sites). Only the few/single-layered 2H-phase MoS₂ with sufficient in-plane S vacancies can catalyze methanol synthesis, while MoS₂ with abundant edge S vacancies mainly promotes methane production.¹ Although the synthesis of MoS₂ is not complicated, conventional synthetic MoS₂ exhibited poor performance and the preparation of highly active MoS₂ catalysts for methanol synthesis from CO₂ hydrogenation is extremely challenging. Previous method for the preparation of few-layer MoS₂ required hydro-solvothermal synthesis in a specially designed airtight autoclave at high temperature and extremely high pressure (400 °C and > 600 bar), imposing tremendous synthetic burden and process risks.¹ Under these harsh reaction conditions, for example, a special pressure-resistant autoclave was required, instead of using common autoclaves which are normally operated at ~ 200 °C and 50 bar.

The first novelty of our work is that, benefiting from our key skills about controlled synthesis of ultrafine MoO₂ nanoparticles and uniformly silica encapsulated structure, the

formed MoS₂ does not aggregate into a multilayer/thicklayer structure during the subsequent mild sulfurization process (~ 200 °C and 50 bar), but rather a 3-4 layer structure (few-layer). The formation of few-layer MoS₂ structures plays a crucial role in efficient methanol synthesis.

(2) We agree with the reviewers' remarks about the fabrication of MoS₂ with strain has been intensively studied. However, most of reported method on the preparation of strained MoS₂ is mainly based on the inheritance from deformed flexible substrates or patterned rigid substrates via mechanical transfer process.² These methods are complicated to adopt and difficult to scale up; they are generally expensive and require specialized equipment.³⁻⁴ In addition, these preparative processes are mainly based on electrochemical applications and the obtained form of MoS₂ usually features multilayer stacks, thus not suitable to high-pressure heterogeneous gas-solid systems. To our knowledge, there are no report of strained MoS₂ with both few-layer and fullerene-like structures.

The second novelty of our work is that, benefiting from the SiO₂-constrained sulfurization process, the formed MoS₂ features both few-layer and biaxially strained fullerene-like structures. The spherical curvature enabled the generation of strain and Sv in the inert MoS₂ basal plane. More importantly, fullerene-like structure of MoS₂ can selectively expose in-plane Sv and reduce the exposure of edge Sv, which were conducive to selective methanol production. For the first time, we applied this well-designed MoS₂ catalysts to methanol synthesis and achieved excellent performance.

1. Hu J, Yu L, Deng J, et al. Sulfur vacancy-rich MoS₂ as a catalyst for the hydrogenation of CO₂ to methanol. *Nat. Catal.*, 2021, 4(3): 242-250.
2. Yang S, Chen Y, Jiang C. Strain engineering of two-dimensional materials: methods, properties, and applications. *InfoMat*, 2021, 3(4): 397-420.
3. You B, Tang M T, Tsai C, et al. Enhancing electrocatalytic water splitting by strain engineering. *Adv. Mater.*, 2019, 31(17): 1807001.
4. Pandey M, Pandey C, Ahuja R, et al. Straining techniques for strain engineering of 2D materials towards flexible straintronic applications. *Nano Energy*, 2023: 108278.

1. It is understood that the SiO₂ shell is employed here to confine the transformation of the MoO₃ core to form FL-MoS₂ hollow spheres. However, the reviewer wonders if it is necessary to keep the SiO₂ shell for catalysis as it may block the active surface and hinder the transport of reactants/products. If the SiO₂ shell was to be removed, does it affect the activity?

Reply: Thank you for the good question. The SiO₂ shell can be easily removed with KOH solution. HRTEM image of obtained MoS₂ (MoS₂-R) showed no fullerene-like structures, but rather randomly aggregated and stacked structure (**Supplementary Fig. 15**). This means that the SiO₂ shell not only confine the transformation of MoO₃ cores to MoS₂ hollow spheres, but also isolates MoS₂ to prevent it from aggregating and maintains its strained, few-layer and fullerene-like structure during the reaction. We also tested the catalytic performance of MoS₂-R samples (**Supplementary Table 4**), which showed significantly lower conversion and methanol selectivity than MoS₂@SiO₂. This result further confirms the importance of strained fullerene-like structure of few-layer MoS₂ with selectively

exposed in-plane Sv for efficient methanol synthesis.

Supplementary Fig. 19. TEM images of MoS₂-R.

Supplementary Table 4. Catalytic performance of different samples in CO₂ hydrogenation to methanol.

Sample names	Reaction conditions	Conversion (%)	MeOH Selectivity (%)	CO Selectivity (%)	CH ₄ Selectivity (%)
MoS ₂ -R	260 °C, 5 MPa,	8.9	44.7	24.8	30.5
MoS ₂ @SiO ₂	GHSV=8000 mL·g _{cat.} ⁻¹ ·h ⁻¹	11.1	52.16	22.5	24.6

MoS₂-R: the SiO₂ shell of MoS₂@SiO₂ were removed.

2. One of the most concerning aspects is the chemical state of Cu and its role in catalysis. The author performed XAS measurements and EXAFS fitting and claimed that (1) Cu is bonded to S and (2) Mo atom is not replaced by Cu in reduced Cu/MoS₂@SiO₂ catalyst. The former is clear from Figure 4e while the latter is not well supported by the experimental data. As the FL-MoS₂ exhibits S vacancies, it can be expected that the CN of Mo is significantly lower than that in the bulk structure. Therefore, the comparison of CN of Cu with that of Mo in bulk MoS₂ is not valid. The author should perform EXAFS fitting for Mo and compare Cu with it to clarify the chemical environment of Cu. The model shown in Figure 4f is not convincing as Cu single atoms located in the MoS₂ lattice could exhibit a similar EXAFS spectrum.

Reply: The synchrotron-radiation-based X-ray absorption spectra (XAS, composed of X-ray absorption near-edge structure (XANES) and the extended X-ray absorption fine structure (EXAFS)) were performed at the XAFCA beamline of the Singapore Synchrotron Light Source (SSLS). Due to the limited X-ray energy range of our XAFCA beamline, we were unable to collect Mo K edge EXAFS data. We agree with you about the coordination number (CN) of Mo is lower than that in the bulk structure. In fact, according to previous literature, the CN of Mo in vacancy-rich MoS₂ approximately equals to 5 (the standard CN of Mo in bulk MoS₂ is 6).¹ Nevertheless, according to the EXAFS fitting results, the CN of Cu–S bond is 1.5 (much lower than the CN of Mo). If Cu replaced the Mo, the CN of Cu–S bond should be close to 5 (or even 6).² We therefore conclude that Cu was not located

in the MoS₂ lattice. In addition, to better explain the position of Cu, we provided a Fourier-transformed EXAFS (FT-EXAFS) spectrum of standard 2H-MoS₂. It is clear that Mo K edge FT-EXAFS of MoS₂ showed both conspicuous Mo-Mo and Mo-S scattering paths. If Cu replace the position of Mo, then we should see a significant Cu-Mo path in Cu K-edge EXAFS spectra.³ From our spectrum, we cannot see significant Cu-Mo peaks, which means that Cu is not in the position of Mo.

EXAFS spectrum of standard 2H-MoS₂

Our co-author of this manuscript, furthermore, Dr. Shibo Xi (responsible for XAS test and data analysis), is a high-level expert in the XAS field. He recently published his work on Ni-doped MoS₂ as a corresponding author in *Nature Nanotechnology*. (<https://doi.org/10.1038/s41565-023-01407-1>).² They employed a one-pot hydrothermal synthesis to prepare single-atom-doped MoS₂ and used Raman spectroscopy and EXAFS fitting to demonstrate that the doped metal is located in the MoS₂ lattice. Their first evidence is that the metal-S CN is close to the CN of Mo (i.e., six). With regards to our Cu/MoS₂@SiO₂, the Cu EXAFS analysis revealed that the CN of Cu-S was found to be 1.5, implying no metal replacement in the lattice. Aside from the comparison of Mo CN, which has high reliability in confirming the position of metal, their second evidence is the red shift of E_{2g}¹ and A_{1g} modes in Raman spectra of metal-doped MoS₂. Therefore, to further confirm whether the Cu was located inside the lattice or dispersed on the surface of MoS₂ in the present study, our prepared samples were also subjected to Raman spectroscopy. As seen from Figure **Supplementary Fig. 19.**, all samples displayed three main characteristic MoS₂ Raman shifts at 383.0, 407.7, and 455.2 cm⁻¹, corresponding to the in-plane Mo-S phonon mode (E_{2g}¹), the out-of-plane Mo-S mode (A_{1g}), and the second-order Raman scattering 2LA(M), respectively. The substitutional replacement of Mo sites by other metal atoms could soften the Mo-S related modes and decrease their vibration frequency, resulting in the red shift of E_{2g}¹ and A_{1g} peaks.^{2, 4} However, unlike the results of previously literature,² the positions of E_{2g}¹ and A_{1g} peaks of the MoS₂@SiO₂ and Cu/MoS₂@SiO₂ are almost identical, indicating that Cu is not located in the lattice of MoS₂.

Supplementary Fig. 19. Raman spectra of MoS₂@SiO₂ and Cu/MoS₂@SiO₂.

1. Hu J, Yu L, Deng J, et al. Sulfur vacancy-rich MoS₂ as a catalyst for the hydrogenation of CO₂ to methanol. *Nat. Catal.*, 2021, 4(3): 242-250.
2. Sun T, Tang Z, Zang W, Xi S, Lu J, et al. Ferromagnetic single-atom spin catalyst for boosting water splitting. *Nat. Nanotechnol.*, 2023, <https://doi.org/10.1038/s41565-023-01407-1>.
3. Jiang K, Luo M, Liu Z, et al. Rational strain engineering of single-atom ruthenium on nanoporous MoS₂ for highly efficient hydrogen evolution. *Nat. Commun.*, 2021, 12(1): 1687.
4. Deng J, Li H, Wang S, et al. Multiscale structural and electronic control of molybdenum disulfide foam for highly efficient hydrogen production. *Nat. Commun.*, 2017, 8(1): 14430.

3. In addition, in the DFT calculations, the author did not take the atomic Cu species into account when constructing the models. It should be clarified if the role of Cu is only to induce a higher number of S vacancies in MoS₂ or if it can optimize the reaction energetics in the CO₂ hydrogenation reactions.

Reply: We are thankful to the Reviewer for bringing this point to us. According to the Reviewer's suggestion, we have incorporated Cu species in our new DFT calculation model and confirmed that Cu single atom promotes the removal of S atoms surrounding the Cu atom to create the active site (adjacent S vacancy pairs) for CO₂ hydrogenation, which is consistent with our previous characterizations. In this revised manuscript, we have demonstrated that Cu atom can easily diffuse around S vacancies, which indicates it can adjust its position to optimize the reaction energetics.

Fig. 6. DFT studies on the role of Cu. **a,b,** Sv formation energies of different S sites when

a Cu atom is anchored at (a) Mo atop, and (b) hollow sites. The arrow indicates the energy value in absence of Cu. c, Cu migration within the two-vacancy site. Inset: atomic structures of the initial, transition, and final states.

We have identified that Mo atop and hollow sites are the most thermodynamically stable sites for Cu single atoms on MoS₂ basal plane (**Supplementary Table 6**). The proximity to Cu dopants decreases Sv formation energies by 1.5 eV (**Fig. 6a, b**). Charge density difference plots indicate a depletion of charge in the region of Mo-S bonds of S atoms coordinated to Cu, which weakens the Mo-S bonds and promotes S removal (**Supplementary Fig. 27**). Following the removal of a neighbouring S atom, Cu fills the resulting vacancy upon the geometry optimization. The formation of the second Sv in this structure is also easier than in pristine MoS₂, especially, next to the Cu dopant (**Supplementary Fig. 28**). The Cu atom exhibits facile migration between these two Sv sites, with a low barrier of 0.33 eV (**Fig. 6c**).

4. The in-situ DRIFT measurements provide limited insights on the benefitting role of Cu as the IR bands intensity of intermediates and CH₃OH are expected to be greater if the catalyst already exhibits higher CH₃OH production rates from CO₂ hydrogenation.

Reply: Currently, there are two main reaction mechanisms for the hydrogenation of carbon dioxide to methanol: HCOO* mechanism and CO* mechanism. Typically, the reaction intermediate for copper-based oxide catalysts is HCOO*. The DRIFTS spectra of Cu/MoS₂@SiO₂ and MoS₂@SiO₂ catalyst showed very similar absorbance bands (CO* was a reaction intermediate and no HCOO* was detected). This suggests that the benefitting role of Cu is only as a promoter. In addition, the IR bands intensity of CO* and CH₃O* over Cu/MoS₂@SiO₂ were much greater than that of MoS₂@SiO₂, which indicates that the formation of CO* from chemisorbed CO₂ and its subsequent hydrogenation to CH₃O* were improved by the incorporation of Cu. This is mainly due to the fact that Cu/MoS₂@SiO₂ have more Sv and activated hydrogen species.

Recently, some researchers have also used in-situ DRIFT to gain insights into the variation of reaction intermediates and study the benefitting role of Cu or Ga.¹⁻² We agree with the Reviewer's point about the limited insights of in-situ DRIFT. To further understand the promoting effect of Cu on catalytic performance, the adsorption and desorption of H₂ on Cu/MoS₂@SiO₂ and MoS₂@SiO₂ were investigated. H₂-TPD experiment was carried out since it could provide valuable information about hydrogen ad-species and their relative concentrations at the surface of catalysts. As shown in **Supplementary Fig. 32**, there is a main peak at around 300 °C for these two catalysts, which can be attributed to the desorption of weakly adsorbed hydrogen-species (probably weak-dissociatively adsorbed H-species).^{1, 3-5} It is clear that Cu/MoS₂@SiO₂ exhibits a higher desorption capacity of H₂, indicating that the higher concentration of H-adspecies of Cu/MoS₂@SiO₂. Therefore, the promoting role of Cu is mainly attributed to facilitate the generation of Sv and dissociation/activation of H₂, based on our studies of EPR, in-situ DRIFT, H₂-TPD and DFT calculations.

Supplementary Fig. 32. H₂-TPD of MoS₂@SiO₂ and Cu/MoS₂@SiO₂.

1. Sha F, Tang C, Tang S, et al. The promoting role of Ga in ZnZrO_x solid solution catalyst for CO₂ hydrogenation to methanol. *J. Catal.*, 2021, 404: 383-392.
2. Xu D, Hong X, Liu G. Highly dispersed metal doping to ZnZr oxide catalyst for CO₂ hydrogenation to methanol: Insight into hydrogen spillover. *J. Catal.*, 2021, 393: 207-214.
3. Li X S, Xin Q, Guo X X, et al. Reversible hydrogen adsorption on MoS₂ studied by temperature-programmed desorption and temperature-programmed reduction. *J. Catal.*, 1992, 137(2): 385-393.
4. Hu X, Qin W, Guan Q, et al. The synergistic effect of CuZnCeO_x in controlling the formation of methanol and CO from CO₂ hydrogenation. *ChemCatChem*, 2018, 10(19): 4438-4449.
5. Afanasiev P, Jobic H. On hydrogen adsorption by nanodispersed MoS₂-based catalysts. *J. Catal.*, 2021, 403: 111-120.

5. The author lists in the reference section are highly inconsistent.

Reply: *Nature Communications* uses standard *Nature* referencing style. All authors should be included in reference lists unless there are six or more, in which case only the first author should be given, followed by 'et al.'. Therefore, for references with more than six authors, we listed only the first author, followed by 'et al.', which would make the references look inconsistent. We have rechecked the author lists in the reference section one by one to ensure they meet *Nature Communications* standards.

REVIEWERS' COMMENTS

Reviewer #1 (Remarks to the Author):

All my questions have been properly answered. I would like to suggest the acceptance of this work for publication.

Reviewer #2 (Remarks to the Author):

In the revised manuscript, the authors have conducted additional experiments, characterization, and DFT calculations. One of the major concerns, which is the chemical environment of Cu single atoms, has been adequately illustrated. Its role in facilitating the formation of S vacancies and H₂ activation is also demonstrated by theoretical investigations and TPD measurements. Therefore, I recommend the publication of the revised manuscript in Nat. Comm.

REVIEWER COMMENTS

Reviewer #1 (Remarks to the Author):

All my questions have been properly answered. I would like to suggest the acceptance of this work for publication.

Reply: We sincerely thank the Reviewer for recognizing our revised manuscript!

Reviewer #2 (Remarks to the Author):

In the revised manuscript, the authors have conducted additional experiments, characterization, and DFT calculations. One of the major concerns, which is the chemical environment of Cu single atoms, has been adequately illustrated. Its role in facilitating the formation of S vacancies and H₂ activation is also demonstrated by theoretical investigations and TPD measurements. Therefore, I recommend the publication of the revised manuscript in Nat. Comm.

Reply: We warmly thank the Reviewer for appreciating our efforts in revising the manuscript.